# Assistance dogs for military veterans with PTSD: A systematic review, meta-analysis, and meta-synthesis

**Sarah C. Leighton**[1]*, **Leanne O. Nieforth**[2], **Marguerite E. O'Haire**[2]

**1** Center for the Human-Animal Bond, Department of Comparative Pathobiology, Purdue University, West Lafayette, IN, United States of America, **2** College of Veterinary Medicine, University of Arizona, Oro Valley, AZ, United States of America

* sleight@purdue.edu

**Data Availability Statement:** All relevant data are within the manuscript and its Supporting Information files.

**Funding:** This publication was made possible with support from the Indiana Clinical and Translational

## Abstract

Psychiatric assistance dogs for military veterans with posttraumatic stress disorder (PTSD) currently make up over 19% of assistance dog partnerships globally. We conducted a systematic review of the literature relating to these partnerships, with specific aims to (1) summarize their characteristics, (2) evaluate the quality of existing evidence, and (3) summarize outcomes. A total of 432 records were independently screened (Cohen's kappa = 0.90). Of these, 41 articles (29 peer-reviewed publications and 12 unpublished dissertations) met inclusion criteria. Data extraction was conducted to address the research aims, including a meta-analysis (quantitative outcomes) and meta-synthesis (qualitative outcomes). All peer-reviewed publications on the topic of psychiatric assistance dogs for veterans with PTSD were published within the last five years. The majority of included articles were quantitative (53%), 41% were qualitative, and 6% employed mixed methods. Mean methodological rigor scores were 80% for peer reviewed articles and 71% for dissertations, where higher scores represent more rigorous methodology. Quantitative articles reported significant improvements in the domains of PTSD severity, mental health, and social health. Impacts on physical health and global quality of life appear inconclusive. Meta-analysis (9 articles) revealed that partnership with an assistance dog had a clinically meaningful, significant, and large effect on PTSD severity scores ($g = -1.129$; $p<0.0001$). Qualitative meta-synthesis identified two third order constructs: (1) Impact on the individual: mental & physical health and (2) Impact beyond the individual: building relationships & connection. This synthesis of increasingly prevalent research on assistance dogs for veterans with PTSD provides support for the impact of this complementary and integrative health intervention on PTSD symptom severity, and signs of meaningful improvements in adjacent domains including mental and social health. Gaps between quantitative and qualitative findings, along with the need to report greater demographic detail, highlight key opportunities for future research.

Sciences Institute which is funded in part by Award Number TL1TR002531 from the National Institutes of Health, National Center for Advancing Translational Sciences, Clinical and Translational Sciences Award (LN). The content is solely the responsibility of the authors and does not necessarily represent the official views of funders. The funders had no role in study design, data collection and analysis, decision to publish, or preparation of the manuscript. https://indianactsi. org/.

**Competing interests:** The authors have declared that no competing interests exist.

## Introduction

In 2019, 6,261 United States veterans died by suicide–nearly double the rate of death from suicide among civilian adults [1]. Existing evidence-based treatments for posttraumatic stress disorder (PTSD), although effective for some, have dropout rates as high as 54% and nonresponse rates as high as 50% [2]. With as many as one in three military veterans diagnosed with PTSD at some point during their lifetime, the need to identify and define effective interventions for this condition is critical [3, 4]. Partnership with a psychiatric assistance dog, a type of assistance dog trained to assist individuals with mental health diagnoses including PTSD, has become increasingly popular among veterans with PTSD [5, 6]. Yet even with this growing popularity, there remains a need for empirical data to evaluate their use [7, 8]. The purpose of this literature review is to systematically identify and evaluate existing evidence on the placement of psychiatric assistance dogs for service members or veterans with PTSD.

An individual may be diagnosed with PTSD after witnessing or experiencing a traumatic event, when certain symptoms persist for longer than one month following the trauma. These can include intrusive symptoms (memories of the event, sleep disturbances, flashbacks, etc.), avoidance behaviors, negative changes in cognition and mood, and reactivity [9]. PTSD is particularly prevalent among military veterans, with approximately 23% of Operation Enduring Freedom/Operation Iraqi Freedom veterans and up to 30.4% of Vietnam War veterans receiving a diagnosis of PTSD compared to 7% among civilians [3, 4, 10]. PTSD is associated with significant increases in rates of substance use, major depression, and suicidality, among other conditions [11, 12]. Suicidality is of particular concern, with 31.6 veterans per 100,000 dying by suicide in 2019 compared to 16.8 per 100,000 civilians [1]. Finally, not only is PTSD pervasive and costly, it also tends to be difficult to treat with dropout and nonresponse rates remaining high even for existing gold standard evidence-based treatments. Given this constellation of issues, it is no surprise that pressure has risen in recent years to identify and develop complementary and integrative treatments to address this crisis.

Military sexual trauma (MST) is a related condition that occurs in the aftermath of sexual assault experienced in connection to military service. MST by definition occurs exclusively in military service members and veterans, and is not formally identified as a separate condition in the Diagnostic and Statistical Manual of Mental Disorders; rather, it is considered a subset of military-connected PTSD alongside combat-related PTSD [9]. Research has found the prevalence of MST within the military population to be somewhere between 20% to 43%, with underreporting and inconsistencies in the criteria and definition of MST contributing to uncertainty [13]. Much like combat-related PTSD, MST-related PTSD is associated with a host of other conditions including depression, substance use disorder, and death by suicide [13]. We emphasize the distinction here because the availability of psychiatric assistance dogs for military service members and veterans with MST-related (as opposed to combat-related) PTSD varies between assistance dog organizations.

Partnership with an assistance dog is a type of animal-assisted intervention (i.e., a form of intervention that includes the presence of an animal as an intentional part of the process) [14]. Assistance dogs, defined in the United States as "dogs that are individually trained to do work or perform tasks for people with disabilities" [15], can help individuals with a wide range of disabilities and may accompany them in public spaces where pet dogs are normally prohibited. These include guide dogs trained to help a person who is blind or visually impaired navigate their environment safely; hearing dogs trained to alert their handler to important sounds in their environment; and assistance dogs trained to assist with mobility, alert to the onset of a medical event, or assist a person with a psychiatric condition such as PTSD (known as psychiatric service dogs) [6, 16]. As of 2018, the number of accredited assistance dog organizations

worldwide stood at 135, representing 16,766 assistance dog teams (i.e., handler-assistance dog pairs). Of these, 19% (over 3,000) are psychiatric assistance dogs placed with military service members and veterans with PTSD, making them the fourth largest category after guide dogs, mobility assistance dogs, and assistance dogs for autism [5]. The actual number of psychiatric assistance dog teams worldwide is likely significantly higher, since psychiatric assistance dogs account for the majority of placements from non-accredited providers [17], and some assistance dog handlers self-train their dogs and do not work with a provider at all.

With the incidence of PTSD on the rise and growing anecdotal reports of the benefits of psychiatric assistance dog placements, the need has emerged to empirically evaluate the efficacy of this potential intervention. In 2010, the U.S. Department of Veterans Affairs (VA) received a legislative mandate as part of Section 1077 of that year's National Defense Authorization Act, directing them to perform a study to better understand the impact of assistance dog placements on the quality of life for veterans with PTSD. The results of this $16M study, summarized in the form of a VA Report published in early 2021, are among a growing body of literature exploring the impact of these placements on health outcomes for military service members and veterans with PTSD [18]. To date, there has only been one review exclusively focused on assistance dogs for PTSD [8]. This review collated the literature on assistance dog placement for veterans with PTSD and found 19 articles on the topic. Methodological shortcomings were emphasized, underscoring the need for further rigorous research. To date, no quantitative meta-analysis nor qualitative meta-synthesis has been conducted on the subject.

Since the past review, the literature has more than tripled in size, identifying the need for a new comprehensive review. This more recently published literature includes the previously-mentioned VA study, the results of which were published and reviewed by the National Academies of Sciences, Engineering, and Medicine. These same results were reported by VA officials in front of Congress in early 2021, directly contributing to the August 24th, 2021 signing into law of the Puppies Assisting Wounded Service Members for Veterans Therapy Act ("PAWS Act"). Given the policy implications and interdisciplinary nature of the topic, a periodic systematic collation process is of the utmost importance. Furthermore, the prior review captured results solely from peer-reviewed journals, potentially missing important grey literature (e.g. dissertations, registered trials, etc.) which is essential to combat the "file drawer" effect (wherein publication bias may result from failure to distribute or publish research with negative results) [19].

With research interest on this topic on the rise, the purpose of this review is to synthesize existing literature, including capturing grey literature and congressionally-mandated research, in order to understand the effect of partnership with a psychiatric assistance dog on PTSD symptoms and quality of life for individuals with military-connected PTSD. Our specific aims are to 1) describe the characteristics of assistance dog placements for military-connected PTSD, 2) assess the methodological rigor of existing research, and 3) summarize the reported outcomes of psychiatric assistance dog placements for military-connected PTSD.

## Materials and methods

### Protocol and eligibility criteria

Prior to conducting the systematic review, the Preferred Reporting Items for Systematic Reviews and Meta-Analyses (PRISMA) guidelines were consulted [20, 21]. To address the "file drawer" effect, we sought to incorporate gray literature including theses and registered trials on Clinicaltrials.gov meeting the pre-defined eligibility criteria. While this study was not pre-registered, all study procedures were established in advance, and the study protocol is available upon request.

*Inclusion criteria consisted of*:

1. Publication in English in a peer-reviewed outlet, a clinical trial registry, or as a thesis.

2. Collection of empirical data.

3. Reporting of results relating to the placement of task-trained assistance animals for military service members or veterans with military-connected PTSD.

*Exclusion criteria consisted of*:

1. Research examining the impact of companion or emotional support animals only.

2. Research on assistance animals placed for a disability or condition other than military-connected PTSD.

3. Research in which the assistance animal handlers were not military service members or veterans.

## Search procedure

A comprehensive search was conducted across a total of 11 databases on September 15th, 2021: ERIC, ProQuest (Dissertations & Theses, PTSD pubs, and Research Library), PsycINFO, PubMed, Scopus, and Web of Science. To increase coverage, the HABRI Central database, Clinicaltrails.gov, and Journal of Veterans Studies were also searched. Search terms included:

1. "service animal" or "service dog" or "assistance animal" or "assistance dog" (including both singular and plural terms) and

2. "posttraumatic stress disorder" or "post-traumatic stress disorder" or "post traumatic stress disorder" or "PTSD".

Exact search syntax was adapted according to each database's search vocabulary (S1 Table). Finally, reference lists of articles meeting criteria for inclusion were reviewed for any additional or missed articles.

## Screening

Following the initial search, articles were imported into Covidence [22], an online platform for the performance of systematic review article screening. Duplicates were automatically identified by Covidence and removed. Articles were screened for inclusion by authors SL and LN first based on title and abstract and subsequently based on full text screening (Cohen's kappa = .90). In case of disagreement, resolution was achieved through discussion. A flow chart of study search and screening is provided in Fig 1.

## Data extraction and evaluation

For all articles, the name of the first author, year of publication, country of publication, university (either of dissertation or of corresponding author), whether it was a peer-reviewed publication or dissertation, and journal (if relevant) were collected. Further data extraction was completed to address our specific aims. First, to describe the characteristics of assistance animal placements for service members or veterans with PTSD the following data were extracted: study population (participant demographic and military participation information), assistance animal characteristics (species, breed, age, origin, training, trained tasks), and assistance animal organization (name, non-profit and accreditation status, assistance animal/veteran pairing process). Second, to assess the methodological rigor the following data were extracted: sample

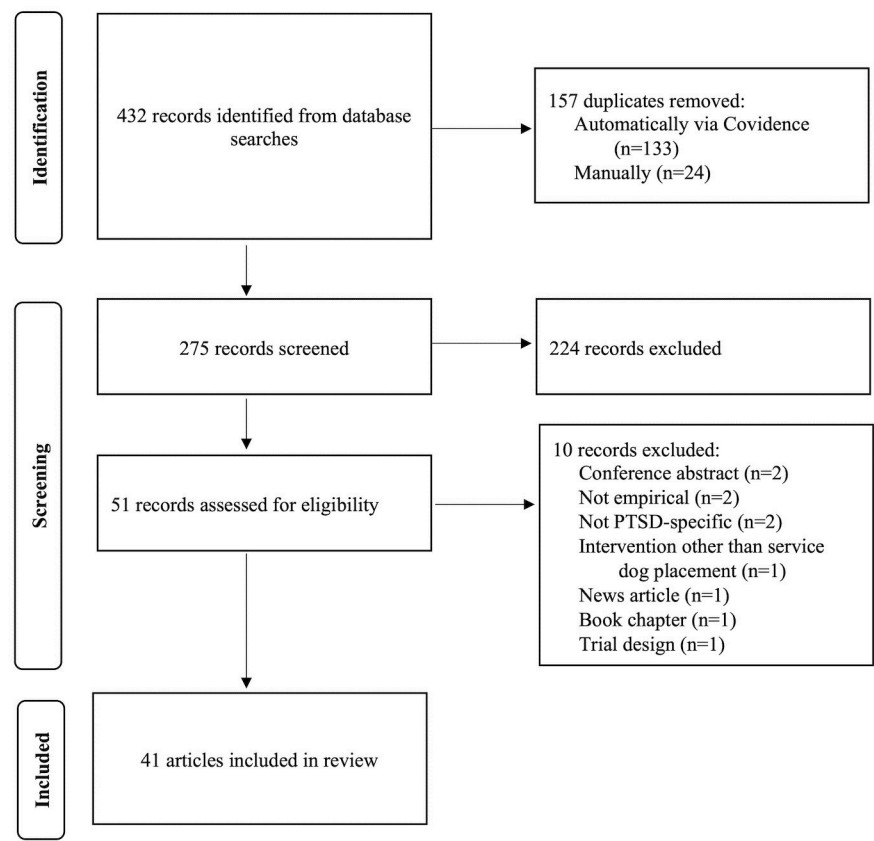

**Fig 1. Study identification and screening for inclusion in systematic review and meta-analysis.**

size, study design, effect size, control condition, ethical approvals, and assessment measures. To assess the quality of the evidence each publication was given a methodological rigor score based on 15 characteristics, replicating the methods used in other systematic reviews in the human-animal interaction field (S2 Table) [23, 24]. Finally, measures and results (including positive, neutral or null, and negative findings) were extracted in order to summarize reported outcomes.

To establish adequate inter-rater reliability authors SL and LN independently coded 20% of the included articles (inter-rater agreement = 92%). All remaining articles were then independently coded by author SL. If additional clarification was needed we contacted the corresponding author on the manuscript in question or in some cases to the assistance dog organization(s).

**Statistical analyses.** Bivariate (Pearson) correlations were used to examine the association between methodological rigor and year of publication or sample size, and an unpaired t-test was used to compare methodological rigor with type of publication (i.e., peer-reviewed or dissertation). For mixed methods articles (n = 2), a single mean of the quantitative and qualitative rigor scores was employed in the statistical analyses to ensure that all articles were weighted equally overall.

To summarize quantitative outcomes a meta-analysis was performed for articles that reported comparable results. Where multiple articles reported results for the same study, data from only a single article (the article reporting results for the largest sample size) was used. To allow comparison between the different versions of the PTSD Checklist (PCL) used among the studies, an additional test-equating step was taken to crosswalk PCL Civilian (PCL-C), Specific

(PCL-S), and Military (PCL-M) version scores to an equivalent PCL for DSM-5 (PCL-5) score based on an established procedure [25, 26]. While this crosswalk has previously only been performed using the PCL-C, the PCL-S and PCL-M have the exact same number of questions with nearly identical wording, allowing for use of the same procedure. After completing the crosswalk we calculated standardized mean differences (SMD) and assessed heterogeneity ($I^2$) using RStudio "esc" and "meta" packages [27]. A random-effects model was chosen to account for heterogeneity between studies. Effect size was calculated using Hedge's g, with small, medium, and large effects defined as g $\geq$ .20, .50, and .80 [28].

**Qualitative meta-synthesis.** To summarize qualitative outcomes, a meta-synthesis was performed following established guidelines [29–31]. Mixed methods articles were not included. Papers were first read in full by authors SL and LN, following which study data were extracted into Microsoft Excel [32]. Study data included both direct quotes from study participants ("first order constructs") and the article authors' identified themes and sub-themes ("second order constructs"). Data analysis was conducted by authors SL and LN who first independently considered second order constructs in juxtaposition with one another, comparing and contrasting themes in order to identify emergent themes and constructs. A descriptive-interpretive approach was employed whereby the authors grouped study data into domains, then further into meaning units, which ultimately were clustered together according to their similarities [31]. Finally, through an iterative discussion-based process, these groupings were refined into final "third order" constructs encompassing the information emergent from the included articles.

## Results

Identification of articles for inclusion in this review is summarized in the PRISMA diagram in Fig 1. A total of 432 records were identified, of which 157 were duplicates removed either automatically via the Covidence system (n = 133) or manually if not recognized by the system (n = 24). An additional 224 records were excluded following title/abstract screening and 10 via full text screening. A total of 41 articles were included in the final sample (29 peer-reviewed publications, 12 unpublished dissertations). Interventions in all 41 articles consisted of placement with an assistance dog as opposed to another species of assistance animal (i.e., miniature horses). All articles referred to participants as veterans, with only one study mentioning active-duty service members being included within the sample [25]. The majority (66%, n = 19/29) of peer-reviewed literature on the topic of assistance dogs and veterans with PTSD was published between 2019–2021, and 100% of peer-reviewed publications on this topic are from the last 5 years (see Fig 2). The majority of articles published were from the United States (94%, n = 32) followed by Canada (18%, n = 6), Australia (3%, n = 1), Denmark (3%, n = 1), and the Netherlands (3%, n = 1).

### Studies with non-veteran participants

Seven articles were excluded from specific aims analyses, as the participants in these studies were not themselves veterans [33–39]. Excluded articles, summarized in Table 1. The participants in these articles were partners and family members, the staff involved in the training and placement of the dog, the mental health professionals providing care, and the assistance dogs themselves. The majority (57%, n = 4) of the seven excluded articles employed a qualitative study design; the remaining studies were cross-sectional (14%, n = 1), repeated measures (14%, n = 1), and mixed measures (14%, n = 1). The average sample size was 27 participants (*SD* = 21.36; range 3–60), with two of the studies having a sample size greater than or equal to 50.

Analyses to address the specific aims were performed for the remaining articles (n = 34) wherein the majority of study participants were veterans with PTSD and the intervention was placement with an assistance dog.

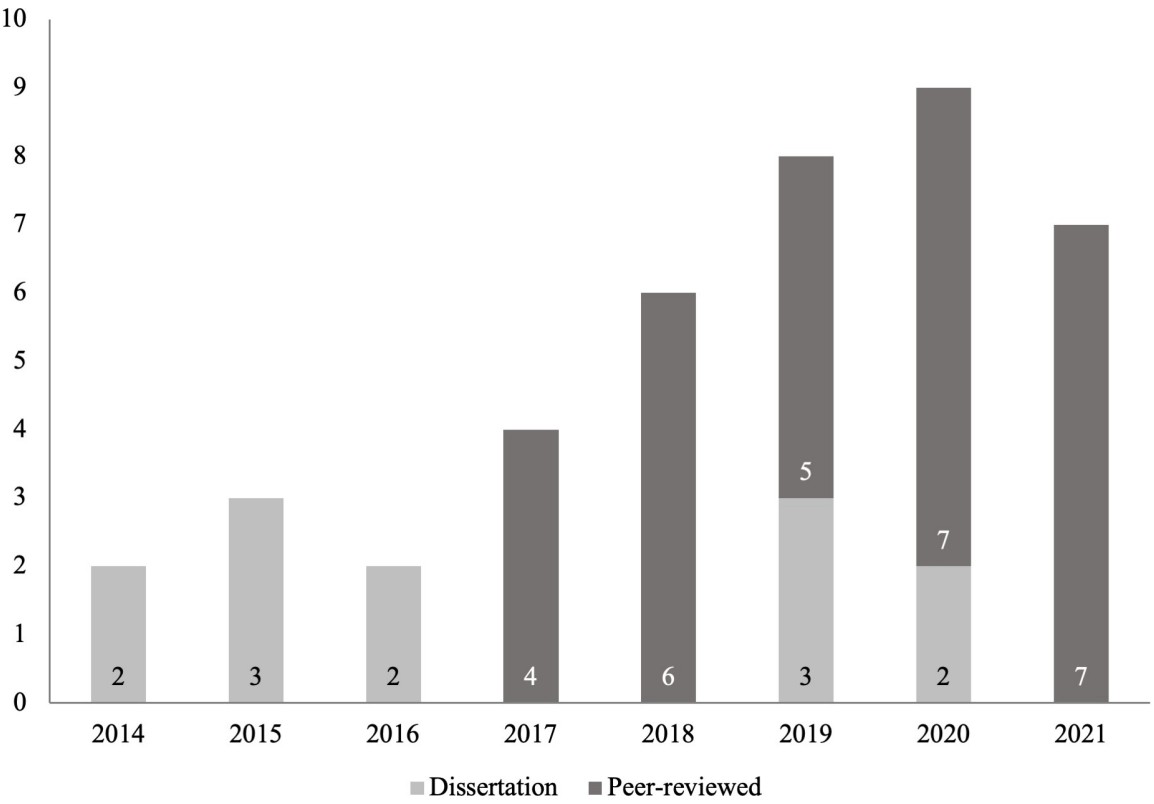

**Fig 2. Dissertations (n = 12) and peer-reviewed publications (n = 29) relating to service dogs and veterans with PTSD by year.**

### Placement characteristics

Our first aim was to describe the characteristics of assistance dog placements for military-connected PTSD. Data were extracted from each of the veteran-specific articles (n = 34) pertaining to the study characteristics (Table 2), participants (Table 3), organizations (S3 Table), and assistance dogs (S4 Table).

**Study characteristics.** Study characteristics are summarized in Table 2. Of the 34 studies, roughly half (53%, n = 18) were quantitative, 41% (n = 14) were qualitative, and 6% (n = 2) employed mixed methods. Quantitative study designs (n = 18) included cross-sectional (50%, n = 9), pre-post (22%, n = 4), non-randomized control (11%, n = 2), randomized control (6%, n = 1), repeated measures (6%, n = 1), and single subject (6%, n = 1). The majority of articles (53%, n = 18) employed a comparison condition, including: veterans on the waitlist to receive an assistance dog (56%, n = 10); pre-post comparison with collection of baseline data prior to receiving the assistance dog (28%, n = 5); veterans who received an emotional support dog (6%, n = 1); veterans who did not have an assistance dog (6%, n = 1); and the presence or absence of a dog, which could be either the assistance dog or an unfamiliar dog (6%, n = 1). The average sample size was 51.91 participants (*SD* = 64.14; range 1–227) and the total sample across all articles was 1,765. Only 38% of articles (n = 13) reported a sample size greater than 30.

**Participant characteristics.** Participant characteristics are summarized in Table 3. For the articles that reported mean participant age (59%, n = 20) the average age of the total sample (1,440 participants) was 41.57 years (range 35.78–50.90 years). For articles that reported participant gender (94%, n = 32) an average of 76% of participants across the total sample (1,322/ 1,735 participants) were male. While across the entire sample female veterans appear to be

**Table 1. Non-veteran participant studies: Participants, design, and outcomes.**

| Study | Participants (n) | Design | Outcomes |
|---|---|---|---|
| **Peer-reviewed** | | | |
| Woodward 2021 | Veterans training SDs, inpatient program (54) | Repeated measures (ABAB) | ✓ Canine's presence associated with ↓ negative, ↑ positive affect; stronger effect with greater PTSD severity.<br>— Negative affect ↓ with time in program; effect of canine's presence on negative affect also ↓ with time. |
| van Houtert 2021 | SDs (19) | Cross-sectional | ✓ Psychiatric service dogs for veterans with PTSD do not appear stressed when training and performing tasks.<br>— Cortisol ↑ after arrival at training center and after 45m of play. |
| Whitworth 2020 | Partners of veterans in SD training program (15) | Qualitative | ✓ Benefits of supportive relationships (human & dog); better partner relationships via ↑ symptom management; partners help maintain participation in program.<br>— Propose attachment theory-based model: new and improved relationships via SD program forms "relational bridge" leading to ↑ resilience.<br>✗ Ongoing relational challenges; limited support for own needs; challenges with public understanding; partners have own trauma; caregiver burden. |
| McCall 2020 | Partners of veterans with SDs (37) and on WL (23) | Cross-sectional + Qualitative | ✓ More benefits than challenges; ↑ resilience & companionship; ↓ anger, social isolation, & work impairment.<br>— No significant differences in functioning between groups.<br>✗ Unwanted attention in public; financial cost; adjusting to changed roles. |
| Vincent 2017a | SD trainers (12)<br>Veterans with SD (1)<br>Veteran advocates (4)<br>Medical professionals (3)<br>Standards board (1) | Qualitative | • Developed theory-oriented logic model: "Sequence of needs and experience that lead PTSD patients to seek a [Psychiatric Service Dog]." |
| **Dissertations** | | | |
| Genbauffe 2020 | Psychotherapists (3) | Qualitative | — All participants have positive attitudes towards pet dogs.<br>✗ Consider use of SD as potential safety behavior; SD may interfere with natural recovery process; concerned with SD fraud/lack of certification. |
| Tilvaldiev 2019 | SD trainers (11)<br>Mental health professionals (6) | Qualitative | • Developed proposed standards for placement of SDs and veterans with PTSD (23 items). |

**Notes.** Ordered in reverse chronological order within each category (peer-reviewed, dissertation). SD service dog. WL waitlist to receive service dog. PCL-5 PTSD Checklist for DSM-5.✓ Positive outcome; — Neutral or null outcome; ✗ Negative outcome; • Theory-based framework or protocol result. ↑ increased; ↓ decreased.

well-represented (24%, compared to 8.2% of veterans in the United States overall) [40], some studies had samples composed of exclusively male veterans (range 43%-100%). Only 3 articles (9%) assessed MST or differentiated between combat- or MST-related PTSD [41–43]. Slightly more than half of articles (53%, n = 18) either reported the United States military branch when describing participants (32%, n = 11) or worked with participants outside of the United States (21%, n = 7). The most common United States military branch represented was the Army (60%; 354/592 participants) followed by the Marine Corps (16%; 94/592), the Navy (14%; 86/592), the Air Force (11%; 6/592), the Coast Guard (8%; 46/592), and the National Guard (4%; 25/592).

The majority of articles (56%, n = 19) did not report participant race or ethnicity. Among the articles that reported detailed data on this (n = 10), the wording of response options varied widely. For the present summary, groups were created based on all response options used in the reported studies. Due to inconsistent terminology across studies, however, and especially among studies that included fewer race and ethnicity response options, some respondents may be counted in categories that do not fully align with how they might have identified if they were given the full range of options reported here. Racial distribution among n = 484 participants was 78% White/Caucasian, 7% Black/African American, 2% Asian/Asian American, 1% Native Hawaiian/Pacific Islander, 2% American Indian/Alaskan Native/Native American/First

**Table 2. Study characteristics.**

| Study | Design | Control | N | PTSD Measure | Outcomes |
|---|---|---|---|---|---|
| **Peer-reviewed** | | | | | |
| Jensen (2021) | Cross-sectional | WL | 186 SD: 112 WL: 74 | PCL-5 ✓ | ✓ ↓ PCL scores. — PCL scores above diagnostic cutoff. No association with time since placement. |
| Nieforth (2021a) | Qualitative | - | 101 SD: 67 P: 34 | CAPS-5 | ✓ ↓ PTSD symptoms; ↑ emotional reserves. Aid communication and emotional regulation within family. Trained tasks ↑ independence. — Recommendations to providers/mental health teams. ✗ ↑ relational load for partner. |
| Nieforth (2021b) | Qualitative | WL | 128 SD: 69 WL: 59 | PCL-C | ✓ ↑ community engagement, QoL; ↓ anxiety. Trained tasks beneficial, and benefit extends beyond tasks via bond. — Gap between expectations and reality. ✗ ↑ challenges in public[a] and when traveling. |
| Rodriguez (2021) | Cross-sectional | WL | 96 SD: 52 WL: 44 | PCL-C | ✓ ↓ PCL scores. More likely to report decrease in medications. — No effect when comparing self-reported medication list. |
| Williamson (2021) | Single subject (AB) + Qualitative | Pre-Post (12m) | 5 | PCL-5 ✓ | ✓ Clinically (not statistically) significant ↓ PCL scores. Trained tasks ↑ independence, ↓ PTSD symptoms. Report ↓ opioid use. — No effect on DUSI-R SU score. ✗ ↑ challenges in public[a]. Frustrations with training/bonding process. |
| Galsgaard (2020) | Single subject (AB) | Pre-Post (16m) | 4 | PCL-C | ✓ ↑ social engagement, sense of agency. ↓ trauma-related intrusion PTSD symptoms. — Impact on overall PTSD symptoms unclear–↓ PCL scores for 2 of 4 participants. |
| Husband (2020) | Qualitative | - | 4 | - | ✓ ↓ substance use, PTSD symptoms. ↓ or no change in prescription medication. — No change in opiate use. |
| Lessard (2020) | Pre-post | Pre-Post (9m) | 18 | PCL-M ✓ | ✓ ↑ moderate physical activity, steps per day (actigraphy). ↑ sleep (self-report). ↓ median PCL score, depression. — No change in sedentary behaviors, sleep (actigraphy). |
| Richerson (2020) | Randomized control | ESA | 227 SD: 114 ES: 113 | CAPS-5 PCL-5 ✓ | ✓ ↓ disability, ↑ QoL for *both* groups. ↓ PCL score, ↓ suicidal behaviors and ideation for SD group. Based on CAPS-5, PTSD absent in 28% of SD group at 15 months (0% absent at baseline). — Unable to reject null hypothesis for primary and most secondary outcomes. ✗ PTSD symptoms worsened from baseline up until pairing. |
| Rodriguez (2020) | Cross-sectional | WL | 217 SD: 134 WL: 83 | PCL-5 ✓ | ✓ ↓ PCL scores. All trained tasks beneficial. Untrained tasks more important on average. — ↓ task usage with time; ↑ task usage with stronger bond. Trained tasks least helpful for amnesia, risk-taking. Gap between expectations and reality. ✗ Tasks least helpful for difficulty remembering trauma, engaging in reckless behavior. |
| Lafollette (2019) | Cross-sectional | - | 111 | PCL-5 ✓ | ✓ Bond is mutually strong. More frequent +R/bond-based training associated with stronger bond. — No association between PTSD severity and bond or SD behavior or character. ✗ More frequent +P training associated with weaker bond. |
| McLaughlin (2019) | Qualitative | - | 7 | - | ✓ ↓ isolation, anxiety, substance use, suicidal ideation. ↑ feeling of safety, sleep quality, emotional regulation. ✗ Financial (veterinary care). Anticipation of grief due to dog lifespan. |
| Scotland-Coogan (2019a) | Qualitative | - | 15 | - | ✓ ↓ anger. ↑ social engagement. Improved relationships. |
| Scotland-Coogan (2019b) | Qualitative | - | 15 | - | ✓ ↓ symptom severity, anxiety, insomnia, nightmares. |
| Whitworth (2019) | Non-randomized control | WL | 30 SD: 15 WL: 15 | TSI-2 | ✓ ↓ PTSD symptoms, depression, anger. Improved relationships. ↑ social engagement. — No change in somatization, understanding/communication, self-care, life activities. |

*(Continued)*

**Table 2.** (Continued)

| Study | Design | Control | N | PTSD Measure | Outcomes |
|---|---|---|---|---|---|
| Crowe (2018a) | Qualitative | - | 9 | - | ✓ Constant, calming presence. ↑ social and community engagement. |
| Crowe (2018b) | Qualitative | - | 6 | - | ✓ ↑ safety (physical), peace of mind, healthy behaviors. Bond exceeds expectations. Improved relationships overall.<br>✗ Lack of acceptance, strained or lost relationships in some cases. |
| Lessard (2018) | Qualitative | - | 10 | - | ✓ Trained tasks beneficial. Companionship. ↓ medication, PTSD symptoms. ↑ social, community engagement.<br>— Recommendations for providers.<br>✗ Stress (veterinary care). ↑ challenges in public[a]. Issues in acquisition process. |
| O'Haire (2018) | Non-randomized control | WL | 141<br>SD: 75<br>WL: 66 | PCL-C | ✓ ↓ PCL score, depression, absenteeism due to health. ↑ QoL, social functioning.<br>— No effect on physical health, employment status. No effect on treatment participation. |
| Rodriguez (2018) | Cross-sectional | WL | 73<br>SD: 45<br>WL: 28 | PCL-C ✓ | ✓ More typical cortisol awakening response. ↓ anxiety, anger, sleep disturbances, substance use.<br>— No effect on sleep quality. No association between PTSD severity and cortisol awakening response. |
| Yarborough (2018) | Qualitative | - | 55<br>SD: 41<br>P: 8<br>T: 6 | Clinician report | ✓ Trained tasks ↓ PTSD symptoms. ↑ social, community engagement. ↓ suicidality, medication use.<br>— Emphasize importance of preparation. Partners may have mixed feelings.<br>✗ Training is stressful and tiring. Benefits take time. ↑ challenges in public[a]. |
| Kloep (2017) | Pre-post | Pre-Post (6m) | 13 | PCL-S ✓ | ✓ ↓ PCL score, depression, anger. ↑ sleep quality, perceived social support, QoL. |
| Vincent (2017b) | Pre-post | Pre-Post (3m) | 15 | PCL-M ✓ | ✓ ↓ PCL score, depression. ↑ sleep quality.<br>— No effect on QoL and social engagement. |
| Yarborough (2017) | Cross-sectional | WL | 78<br>SD: 24<br>WL: 54 | PCL-C ✓ | ✓ Tasks beneficial. ↓ PCL score, depression. ↑ happiness, QoL. Improved mental health overall.<br>— No effect on physical health, activity level. |
| **Dissertations** | | | | | |
| Floore-Guetschow (2020) | Qualitative | - | 7 | - | ✓ ↑ enjoyment of day-to-day life, social and community engagement. ↓ PTSD symptoms.<br>— Some re-engaged/continued mental health treatments; others discontinued.<br>✗ Anticipation of grief due to dog lifespan. ↑ challenges in public[a]. |
| Hansen (2019) | Cross-sectional | Combat vs. non-combat | 64 | PCL-5 | ✓ Task usage positively associated with PTSD severity.<br>— Insecure attachment style positively associated with task usage. |
| Parenti (2019) | Repeated measures (ABCDE) | Dog (SD / pet) present / absent | 6 | PCL-5 | ✓ ↓ heart rate, negative affect in presence of dog. Negative affect significantly lower for SD condition.<br>— Stress indicators declined overall throughout session. No effect on heart rate, mental workload. |
| Kegel (2016) | Cross-sectional | V | 66<br>SD: 43<br>V: 23 | PCL-M | ✓ ↑ QoL (social engagement, creative expression).<br>— No association with PCL score, substance use, other QoL areas.<br>✗ ↑ difficulty falling and staying asleep. |
| Kopicki (2016) | Cross-sectional | WL | 18<br>SD:12<br>WL: 6 | PCL-M | ✓ ↓ PCL (hyperarousal subscale).<br>— No effect on other PCL areas. Placement length not associated with PTSD symptoms, sleep quality. |
| Brown (2015) | Qualitative | - | 1 | - | ✓ ↑ sense of physical and emotional safety. ↓ medication, PTSD symptoms. Tasks beneficial.<br>✗ ↑ challenges in public[a]. |
| Hyde (2015) | Cross-sectional + Qualitative | - | 7 | PCL-5 | ✓ ↑ hope, social and community engagement, routine.<br>— No effect on PCL scores.<br>✗ ↑ challenges in public[a]. Burden of care. Little planning around retirement. |
| Marston (2015) | Cross-sectional | WL | 18<br>SD:12<br>WL: 6 | - | — No effect on QoL. No association with time since pairing. |

(*Continued*)

 

**Table 2.** (Continued)

| Study | Design | Control | N | PTSD Measure | Outcomes |
|---|---|---|---|---|---|
| Moore (2014) | Qualitative | - | 8 | - | ✓ ↓ PTSD symptoms. ↑ sense of safety, social engagement. Trained tasks, bond both beneficial.<br>— Some participants integrated SD in therapy.<br>✗ ↑ challenges in public[a]. |
| Newton (2014) | Qualitative | - | 6 | - | ✓ ↓ PTSD symptoms. ↑ community and social engagement.<br>— Emphasize importance of working with a good organization.<br>✗ ↑ challenges in public[a]. Frustrations with training experiences. |

**Notes.** Ordered by most recent to least recent within each group. SD Veterans with service dogs. WL Waitlist to receive service dog. ES Emotional support dog. Pre Pre-placement. V Veterans not on waitlist nor partnered with service dog. P Partner. T Service dog trainer. PCL PTSD Checklist (-C Civilian; -M Military; -S Specific; -5 for DSM-5). CAPS Clinician-Administered PTSD Scale. TSI-2 Trauma Symptom Inventory-2. PCL ✓ Clinically significant improvement in mean score; PCL Mean score change not clinically significant; Italics indicates time points. No symbol: Mean score change not reported. ✓ Positive outcome; — Neutral or null outcome; ✗ Negative outcome. ↑ increased; ↓ decreased. QoL Quality of Life. DUSI-R SU Drug Use Screening Inventory Revised Substance Use Subscale. +R Positive reinforcement. +P Positive punishment.

[a]Challenges in public may include access issues, stigma, and unwanted attention.

Nations, 7% multiple races, 2% other (unspecified), and 2% unknown/decline to answer. One study additionally provided the response option of "Arab," which was selected by no participants. Among 8 studies (n = 318 participants) reporting on ethnicity, 13% of participants identified as Hispanic/Latinx.

The majority of articles (56%, n = 19) did not describe or assess comorbid conditions beyond PTSD. Of those that did (n = 15) the most common other condition assessed was traumatic brain injury (73%, n = 11) followed by comorbid physical disabilities (27%, n = 4), psychiatric diagnoses (20%, n = 3), and substance use disorder (13%, n = 2). More than one third of articles (44%, n = 15) omitted either the percentage of participants concurrently receiving treatment, the percentage taking prescription medication for PTSD, or both. Of the 19 articles that did describe this information, six (32%, n = 6) reported that participants were receiving treatment as usual (TAU) but did not provide further detail. A total of 11 articles reported detailed treatment information: overall, an average of 89% of veterans across the total sample (558 of 625 participants; range 79%-100%) were receiving concurrent treatment. Meanwhile, among the 8 articles that reported detailed prescription medication information, an average of 76% of veterans (217 of 287; range 64%-100%) were receiving prescription medication. The vast majority of studies did not account for other treatment modalities as confounding factors; however, some indicated equivalence across groups at baseline (12%, n = 4), uniform participation in psychiatric treatment as an inclusion requirement for the study (6%, n = 2) or the assistance dog program (12%, n = 4), or reported it as an outcome (n = 3 quantitative, n = 2 qualitative).

**Organization characteristics.** Organizational characteristics are summarized in S3 Table. Across all articles 19 unique organizations were represented with K9s For Warriors being mentioned the most often (24%, n = 8). Over one third (35%, n = 12) of articles did not name the provider(s) involved in the assistance dog placement. While handlers with owner-trained assistance dogs (i.e., who did not work with an organization) were included in the samples for two studies [44, 45], no articles focused on or reported results exclusive to these types of placements. Organizations were nonprofit where specified; 44% of articles (n = 15) did not report this information. Most articles (79%, n = 27) did not report organization accreditation status; of the remaining seven articles, organizations were either accredited through Assistance Dogs

**Table 3. Participant characteristics.**

| Study | Age, M | % Male | Secondary Disabilities Assessed | | | | Medications and Treatments | United States Military Branch (%)[a] | | | | | |
|---|---|---|---|---|---|---|---|---|---|---|---|---|---|
| | | | TBI | SUD | Psy | Phy | | A | AF | CG | MC | N | NG |
| **Peer-reviewed** | | | | | | | | | | | | | |
| Jensen 2021 | 40 | 74 | | | - | | TAU | | | | - | | |
| Nieforth 2021a | - | 79 | | | - | | TAU | | | | - | | |
| Nieforth 2021b | 38 | 80 | ✓ | | | | TAU | 65 | 12 | 0 | 12 | 12 | 0 |
| Rodriguez 2021 | 39 | 78 | | | - | | M: 74%; T: 82% | | | | - | | |
| Williamson 2021 | 43 | 100 | | ✓ | | | M: 100% T: TAU | | | Not USA | | | |
| Galsgaard 2020 | 48 | 100 | | | ✓ | | M:—T: 100%[s] | | | Not USA | | | |
| Husband 2020 | - | 75 | | | - | | M: 100%; T: - | | | Not USA | | | |
| Lessard 2020 | 50 | 83 | | | - | | M: -; T: 88% | | | Not USA | | | |
| Richerson 2020 | 50 | 78 | | | ✓ | ✓ | M: -; T: 100%[s] | 53 | 10 | 2 | 20 | 16 | 11 |
| Rodriguez 2020 | 40 | 75 | | | - | | TAU | | | | - | | |
| Lafollette 2019 | 40 | 80 | | | - | | - | | | | - | | |
| McLaughlin 2019 | - | 86 | | | - | | - | | | Not USA | | | |
| Scotland-Coogan 2019a | - | - | | | - | | - | | | | - | | |
| Scotland-Coogan 2019b | - | - | | | - | | - | | | | - | | |
| Whitworth 2019 | 51 | 87 | | | - | | M: 93%; T: 90% | 57 | 3 | 3 | 13 | 23 | 0 |
| Crowe 2018a | 36 | 100 | | | - | | M: -; T: 100%[P] | 56 | 11 | 0 | 33 | 11 | 0 |
| Crowe 2018b | 43 | 67 | ✓ | | | | M: 100%; T: 100%[P] | 83 | 17 | 0 | 17 | 0 | 0 |
| Lessard 2018 | - | 90 | | | - | | - | | | Not USA | | | |
| O'Haire 2018 | 37 | 78 | ✓ | | | ✓ | M: TAU; T: 79% | 66 | 10 | 0 | 11 | 13 | 0 |
| Rodriguez 2018 | 37 | 81 | ✓ | | | | M: -; T: 79% | 56 | 12 | 0 | 12 | 19 | 0 |
| Yarborough 2018 | 45 | 68 | | | - | | M: 64%; T: - | | | | - | | |
| Kloep 2017 | 38 | 69 | ✓ | | ✓ | | M: 100%; T: 100%[P] | | | | - | | |
| Vincent 2017b | - | 75 | | | - | | - | | | Not USA | | | |
| Yarborough 2017 | 42 | 69 | | | - | | M: 70% T: - | | | | - | | |
| **Dissertations** | | | | | | | | | | | | | |
| Floore-Guetschow 2020 | - | 43 | | | - | | - | 43 | 14 | 0 | 29 | 29 | 0 |
| Hansen 2019 | - | 69 | | | | ✓ | - | | | | - | | |
| Parenti 2019 | - | 100 | | | - | | - | | | | - | | |
| Kegel 2016 | - | 58 | ✓ | ✓ | | | - | | | | - | | |
| Kopicki 2016 | 39 | 74 | ✓ | | | | - | 65 | 25 | 0 | 20 | 10 | 0 |
| Brown 2015 | - | 100 | | | - | | - | | | | - | | |
| Hyde 2015 | 44 | 100 | ✓ | | | | - | | | | - | | |
| Marston 2015 | 39 | 74 | ✓ | | | | - | 65 | 25 | 0 | 20 | 10 | 0 |
| Moore 2014 | - | 75 | ✓ | | | ✓ | M: -; T: 100%[P] | 50 | 13 | 0 | 13 | 25 | 0 |
| Newton 2014 | - | 83 | ✓ | | | | - | | | | - | | |

**Notes.** Ordered by most to least recent within each group.—Not reported. TBI Traumatic Brain Injury. SUD Substance use disorder. Psy Psychiatric disability (other than PTSD). Phy Physical disability. M Medication. T Treatment. TAU Treatment as usual. A Army. AF Air Force. CG Coast Guard. MC Marine Corps. N Navy. NG National Guard.

[s]Requirement of study.

[P]Requirement of program.

[a]May exceed 100% if participants served with multiple branches.

International (ADI), adherent to ADI standards, or in one case were developing their own accreditation system. Nearly half of the articles (47%, n = 16) did not report on the format for

partnering veterans with assistance dog. Of the 18 articles reporting this information the most common format (56%, n = 10) employed by organizations was an immersive format (i.e., onsite classes with multiple participants, generally requiring overnight stays) with an average duration of 2.65 weeks (*SD* = .58; range 1–3 weeks). The next most common format (39%, n = 7) consisted of the veteran training the assistance dog themselves with guidance from the organization's staff over an average of 29.07 weeks (*SD* = 16.95; range 14–78 weeks). One article (6%, n = 1) had multiple organizations whose partnering formats varied represented within their sample. Finally, the majority of articles did not report veteran and assistance dog team certification standards (68%, n = 23); of the remaining 11 articles, the most common team certification took place through ADI public access testing (45%, n = 5).

**Psychiatric assistance animal characteristics.** Assistance dog characteristics are summarized in S4 Table. The assistance animals were dogs for 100% of studies. The majority of articles (74%, n = 25) did not report the assistance dog breeds; of the nine that did the breeds represented were Labrador Retrievers (78%, n = 7), Golden Retrievers (67%, n = 6), Labrador Retriever mixes (44%, n = 4), German Shepherd Dogs (33%, n = 3), mixed breeds (22%, n = 2), and "other" (22%, n = 2). Only two articles reported the assistance dog's age at the time of placement: assistance dogs were 2 years old at the time of placement in one study [46], and in the other study veterans received dogs as puppies and trained them with guidance from the organization over the course of 10 months [47]. The majority of articles (59%, n = 20) did not report dog origin. Of the 14 that did the majority were from shelters (71%, n = 10). Other origins included assistance-dog specific breeding programs (referred to as "purpose-bred"; 29%, n = 4), rescues (21%, n = 3), agency transfers (i.e., received from another assistance dog school; 7%, n = 1), and other sources (7%, n = 1). Training format varied across the 19 articles reporting this characteristic including training by organizational staff (58%, n = 11), by the veteran with guidance from the organization (37%, n = 7), or by persons who are incarcerated in a correctional facility (11%, n = 2); 44% of articles (n = 15) did not report this information. Length of time in training was reported in a variety of ways with some articles reporting hours of training (ranging from 60+ to 900+ hours) and others reporting the duration of the training period in weeks or months (ranging from 14 weeks to 2 years); the majority of articles (65%, n = 22) did not report length of training. Finally, of the articles that gave specific examples of PTSD-related trained tasks (Fig 3; n = 18), the most common tasks mentioned related to creating space to increase comfort in public (78%, n = 14). More than half of articles also mentioned tasks relating to anxiety interruption (67%, n = 12), monitoring the environment (56%, n = 10), and nightmare interruption (56%, n = 10). Over one third (38%, n = 13) of articles did not mention trained tasks at all.

## Methodological rigor

Our second aim was to assess the methodological rigor of existing research. A set of 15 methodological criteria were rated on a binary scale (0 = no, 1 = yes) and summed for each article (detailed in S2 Table), resulting in a final percentage score. The mean rigor score for the non-veteran specific studies (Table 1) was 87% (range 69%-100%). For the studies with veteran participants the average rigor score for peer-reviewed studies was 80% (range 53%-100%) compared to 71% for dissertations (range 47%-92%). Methodological rigor was not significantly correlated with year of publication (*r* = .247, *p* = .204) or type of article (i.e., peer-reviewed vs. dissertation; *t* = 1.599, *p* = .128). However, there was a significant positive correlation between methodological rigor and sample size (*r* = .530, *p* = .001).

**Quantitative studies.** The number of articles with quantitative components (n = 20) meeting each rigor criterion is summarized in Fig 4. All 20 articles clearly stated their aim or

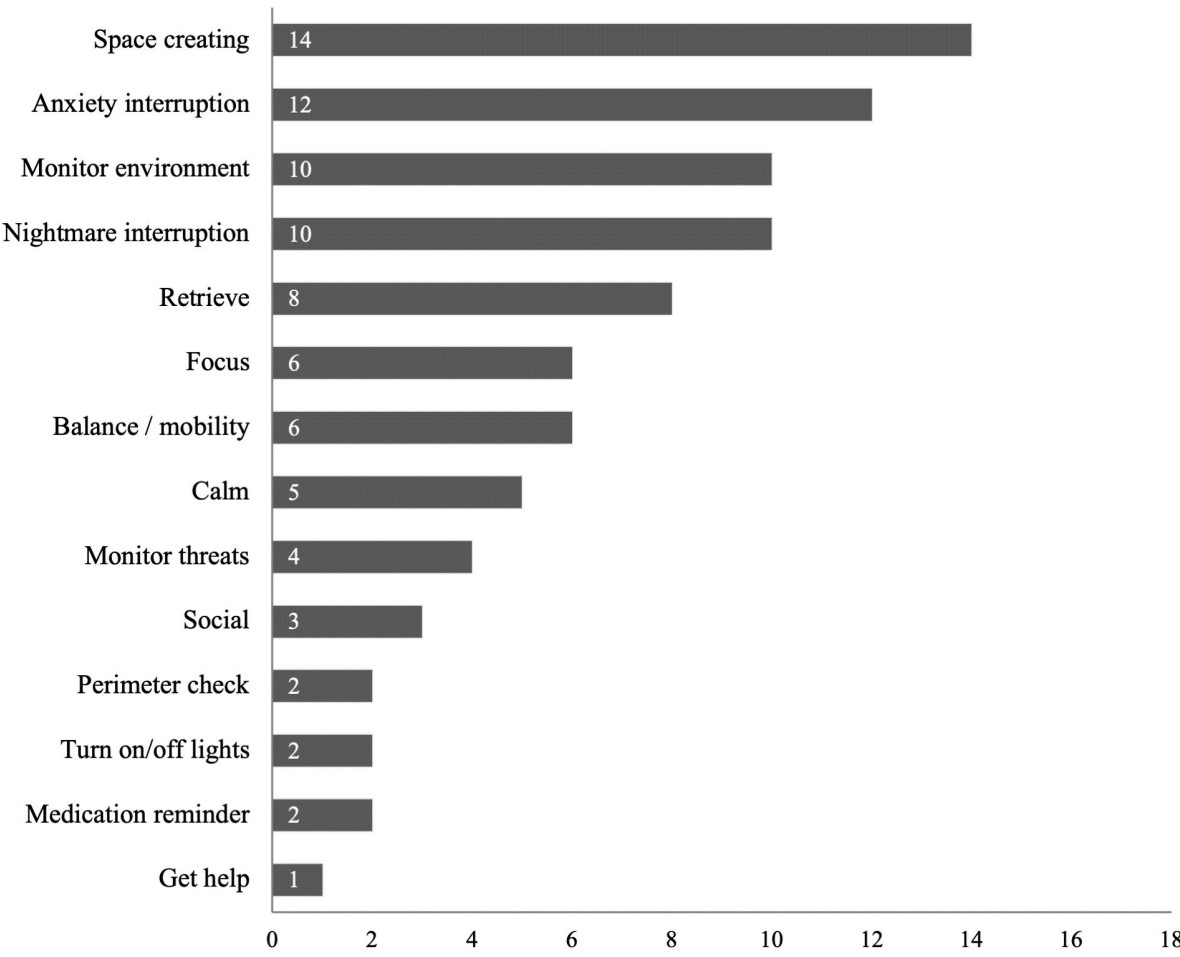

**Fig 3. Number of studies (n = 18) reporting PTSD-specific trained tasks.**

purpose. Half or fewer articles described placement characteristics (50%, n = 10) or reported effect sizes (45%, n = 9). Among studies with quantitative components, the mean rigor score was 84% (range 53%-100%) for peer-reviewed publications and 73% (range 47%-92%) for dissertations.

**Qualitative studies.** The number of articles with qualitative components (n = 16) meeting each rigor criterion is summarized in Fig 5. All 16 articles clearly stated their aim or purpose, provided sequences of original data (e.g., quotations), clearly explained their methods, and gave plausible and coherent explanations for their results. Half or fewer articles considered time since placement as a factor when interpreting results (50%, n = 8), employed data triangulation (50%, n = 8), described the characteristics of the assistance dog placements including provider and training (50%, n = 8), described participant's disabilities beyond PTSD or independently assessed PTSD diagnoses (31%, n = 5), or reported achieving data saturation (13%, n = 2). Among studies with qualitative components, the mean rigor score was 76% (range 53%-100%) for peer-reviewed publications and 69% (range 60%-87%) for dissertations.

## Outcomes

Our third and final aim was to summarize the reported outcomes of psychiatric assistance dog placements for military-connected PTSD (Table 2).

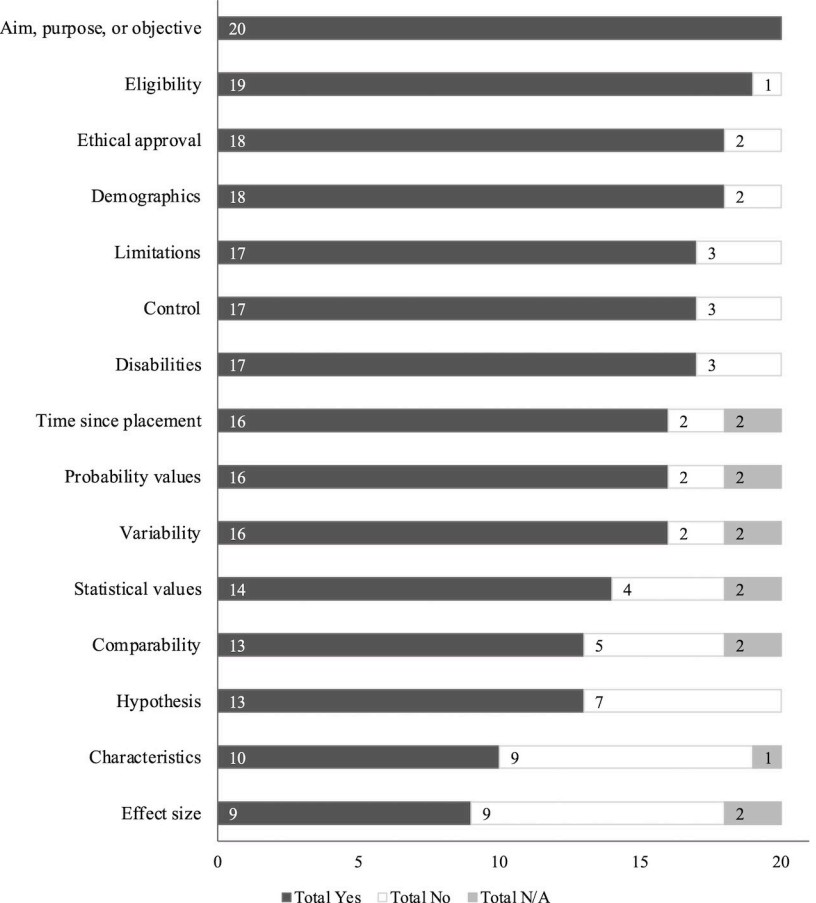

**Fig 4. Number of quantitative studies (n = 20) meeting each rigor criterion, where applicable.**

**Quantitative articles.** *Measures.* Among the 20 articles with quantitative components, all used one or more self-report assessment measure (i.e., standardized survey instrument and/or subscale, or non-standardized survey designed for use specifically by assistance dog handlers), three (15%) used an objective assessment measure (i.e., electrocardiogram, actigraphy, and cortisol), and one (5%) used a clinical assessment measure (i.e., the Clinician-Administered PTSD Scale for DSM-5 [CAPS-5]). A total of 47 different self-report measures were reported across all articles. By far the most common measure used was the PTSD Checklist (PCL), reported in 90% of quantitative articles (n = 18). With the exception of the PCL, the same self-report measures were otherwise rarely repeated across studies. Other repeat measures included the Pittsburgh Sleep Quality Index (PSQI; 25%, n = 5), the Veterans RAND 12-Item Health Survey (VR-12; 20%, n = 4), the Dimensions of Anger Reactions (DAR-5; 15%, n = 3), the Beck Depression Inventory (BDI-2), the Life Space Assessment (LSA), the Patient Health Questionnaire-9 (PHQ-9), and the World Health Organization-Five Well Being Index (WHO-5; each 10%, n = 2). This does not include measures that were repeated across multiple articles reporting outcomes for the same study. None of the objective assessment measures (i.e., electrocardiogram, actigraphy, and cortisol) were repeated across more than one study.

*PTSD severity.* The majority of quantitative articles (80%, n = 16) reported outcomes relating to PTSD severity. Of these articles all but one (94%, n = 15) reported statistically significant positive results, i.e., that assistance dog interventions reduce PTSD symptom severity in one or

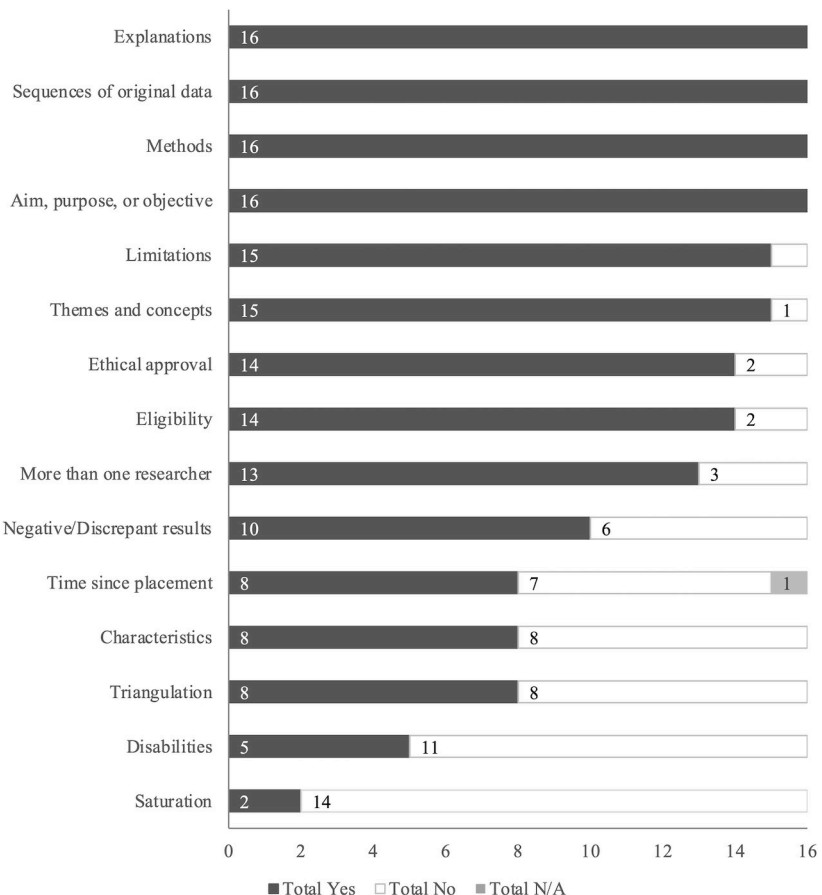

**Fig 5. Number of qualitative studies (n = 16) meeting each rigor criterion, where applicable.**

more domains [6, 18, 25, 41, 43, 47–54]. Given the repeat use of the PCL we performed a meta-analysis to better estimate the magnitude of this effect; the other self-report PTSD measure employed was the Trauma Symptom Inventory-2 (TSI-2), used in one study [52].

The sole randomized controlled trial (RCT, conducted by the VA), which was also the only study to report PTSD severity findings for a clinical measure (i.e., the CAPS-5), found no significant between-group difference for veterans paired with assistance dogs compared to emotional support dogs; however, 32% of the assistance dog group and 26% of the control group no longer met clinical criteria for PTSD at 15 months [18]. No studies differentiated outcomes based on PTSD origin (i.e., combat- or MST-related).

*Quality of life—global.* Most quantitative articles (78%, n = 14) reported outcomes relating to quality of life either globally (39%, n = 7) or for one or more sub-domain including physical health (44%, n = 8), mental health (56%, n = 10), and social health (22%, n = 4). Outcomes for quality of life among the seven articles using global measures were mixed. Three articles reported significant improvement in quality of life globally as measured by the WHO-5, the Quality of Life Scale (QOLS), and the Wisconson Quality of Life Index [41, 47, 53]. In contrast, one dissertation employing the QOLS reported finding some significant positive changes in individual quality of life areas, but that the impact on overall quality of life was not significant [55]. One analysis [18] found no significant between-group difference as measured by the WHO Disability Assessment Schedule (WHODAS 2.0), however they did demonstrate within-group improvement for both groups post-pairing. Finally, two articles reported exclusively

null results for global quality of life as measured by the Brief WHO Quality of Life questionnaire (WHOQOL BREF) and the Quality of Life Index (QLI) [51, 56].

*Physical health*. The physical health subdomain included physical well-being, sleep, activity level, and substance use. A majority of the eight articles assessing physical health outcomes reported null findings (88%, n = 7). Notably, all three studies that reported results using the VR-12 found no effect when analyzing the physical health component (VR-12 PCS) [18, 43, 53].

One article used actigraphy as an objective measure alongside two self-report measures, the LSA and PSQI. While they reported significant positive findings for activity and sleep quality based on both self-report measures, there were no significant actigraphy findings related to sleep [49]. With regard to sleep quality several other articles employing the PSQI self-report measure similarly reported positive findings [18, 51]. One article reported a non-significant trend in the hypothesized direction, but reported significant positive findings for the PROMIS Sleep Disturbance measure [57]. Another reported null findings for the PSQI along with a significant increase in difficulty falling and staying asleep for the assistance dog group based on analysis of a single sleep-related item from the PCL [55].

Finally, three articles quantitatively assessed the impact of assistance dog placement on substance use; one reported significant positive findings [57] while two found no effect [54, 55].

*Mental health—other*. We considered the mental health subdomain to include outcomes for any area relating to psychological well-being other than PTSD severity, described separately. All ten articles reported positive outcomes for mental health measures; more than half (60%, n = 6) reported additional null findings. In contrast to the physical health outcomes, all three articles reporting results for the VR-12 mental health component (VR-12 MCS) found psychiatric assistance dog placement to have a significant positive effect [18, 43, 53]. These findings appear to be in line with positive effects measured through other broad assessments of mental health, namely the Bradburn Scale of Psychological Wellbeing (BSPW), Connor-Davidson Resilience Scale (CDRS), and Satisfaction With Life Scale (SWLS) [43].

Two studies found significant positive impacts on stress as assessed through objective measures. Using physiological measurement (i.e., a cardiography device) one study found a lower average heart rate in the presence of a dog, in combination with reduced negative affect as measured through the PANAS Negative Affect subscale [58]. Another study measured salivary cortisol and found that the assistance dog intervention group had a more typical cortisol awakening response compared to the control group, a possible indicator of improved well-being through the hormonal stress-response system [57].

Six articles found a significant positive effect on depression as measured by the BDI-2, PHQ-9, PROMIS Depression, Quick Inventory of Depressive Symptomatology (QIDS), and BASIS depression subscales [18, 41, 43, 49, 51, 53]. Anger also appears to be positively influenced by psychiatric assistance dog placements as measured through the DAR and PROMIS Anger assessments [18, 41, 43, 57]. Other significant positive mental health impacts included reduced anxiety (PROMIS Anxiety) and increased happiness (General Social Survey) [53, 57].

In spite of the often-cited concern around increased suicidality in veterans with PTSD [1] only one study specifically assessed this mental health domain, finding significant positive within-group impacts for veterans paired with a psychiatric assistance dog [18].

Finally, three articles reported specific information regarding the association between assistance dog partnership and mental health treatment participation. No difference in participation was found between groups, but the assistance dog group perceived greater improvement for a given level of treatment compared to the waitlist control [43]. Veterans with assistance dogs were found to be more likely to be taking psychiatric medications compared to veterans without assistance dogs [55] and were also more likely to self-report decreased dosages, although there was no overall effect on medication use [50].

*Social health.* We considered social health outcomes to include social interactions and relationships as well as community engagement. Though there is evidence for a significant association between PTSD and social isolation, only 22% (n = 4) of quantitative articles specifically assessed this domain. In summary, significant positive findings related to areas of perceived social support, social and societal participation, and companionship as measured by subdomains of the WHODAS 2.0, PROMIS Ability to participate in social activities, PROMIS Social isolation, PROMIS Companionship, and Perceived Social Support Scale (PSSS) [41, 43, 52]. One article also assessed the impact of psychiatric assistance dogs on employment, reporting a significantly lower proportion of health-related absenteeism along with null findings for level of employment and level of at-work impairment as measured through the Work Productivity and Activity Impairment Questionnaire (WPAI) [43].

*Assistance dog partnership.* A small number of articles (28%, n = 5) reported findings specific to the assistance dog partnership. These related to four themes: the assistance dog's working role, the strength of the bond between the handler and dog, the assistance dog's characteristics, and attachment styles between assistance dog and handler.

Three studies investigated task usage. Trained tasks were found to be most helpful for managing the PTSD symptoms of hypervigilance, unwanted upsetting memories, heightened startle reactions, and physical reactivity after exposure to traumatic reminders. On the other hand, tasks were least helpful for PTSD symptoms such as inability to recall key features of the trauma, and participation in risky or destructive behavior [6]. While they found no association between symptom severity and frequency of task use, this is in contrast with another study which found a positive association between task use and PTSD severity [59]. Two articles reported results for task importance, both finding calming, anxiety (or panic) interruption, and space creation to be among the most important [6, 53]; one of these [6] additionally reported environmental and threat monitoring along with nightmare interruption to be of high importance, and that untrained behaviors were more important on average than trained tasks–in contrast with veterans' expectations prior to placement with an assistance dog.

Three studies examined the strength of the human-animal bond between handlers and assistance dogs and its associations. Overall these human-canine partners appear highly and mutually bonded, evidenced through both a standardized self-report measure (the Inclusion of Self in Other Scale [IOS]) and assessment of the assistance dog's attachment-related behaviors [60]. Bond strength was positively associated with the previously-discussed importance of tasks and untrained behaviors, but not with PTSD severity [6, 60]. One study also explored the handler's use of different training techniques, finding that more frequent use of positive reinforcement and/or bond-based training methods were associated with a stronger bond, while more frequent use of positive punishment was associated with a weaker bond [60].

A single study examined the characteristics of psychiatric assistance dogs placed with veterans with PTSD, finding that they tend to be highly food motivated, handler-focused, and interactive, and that the dog's behavior and characteristics were not associated with PTSD severity [60].

Finally, two dissertations examined the assistance dog partnership through the lens of attachment styles, finding that attachment anxiety was associated with partnership of an assistance dog in general [59] and that alignment in degree of assertiveness between the handler and the assistance dog (as opposed to one party being highly assertive and the other being less assertive) was associated with faster recovery from stress reactions [58].

*Meta-analysis.* A meta-analysis was performed for articles that reported results using the PCL. Studies were only included in this meta-analysis if they reported results (including providing sample size, mean, and standard deviation) relative to a "no dog" comparison condition, i.e., at baseline prior to assistance dog placement or for veterans on a waitlist to receive an

assistance dog. Where articles reported scores at multiple time points, the scores for the earliest "no dog" time point and latest "dog" time point were selected (range for "dog" condition: 3–22 months). After pooling studies that met these criteria (n = 9) we found that placement with a psychiatric assistance dog had a significant and large effect on PCL scores ($g = -1.137$, 95% CI: $-1.476$ to $-.796$, $p < .0001$; Fig 6). The weighted average change in means of $-15.13$ points (range $-10$ to $-37$) exceeds the 10 point minimum threshold for clinically significant progress (Table 4) [61]. However, heterogeneity was high between studies ($I^2 = 64\%$) and only one study employed a RCT design [18]. Results for 92% of the total sample (508 of 551 unique participants) originate from studies with a very high methodological rigor score (range 93%-100%).

**Qualitative articles.** *Meta-synthesis.* A qualitative meta-synthesis was performed for the 16 articles employing qualitative methods. First and second order constructs (defined as direct quotes from study participants and the themes identified by authors, respectively) were extracted. After completing the meta-synthesis, two core third-order constructs emerged:

1. *Impact on the individual*: *mental & physical health*.

2. *Impact beyond the individual*: *building relationship & connection*.

*1. Impact on the individual*: *mental & physical health*

Analysis of first and second order constructs identified the first theme of "impact on the individual," which was present in all articles. First and second order constructs, primarily drawn directly from the self-reported experiences of the veterans themselves, appeared supported by the second party observations from their significant others [62, 63].

The impact of a psychiatric assistance dog was most evident in the domain of mental health. Participants in every article spoke about decreases in PTSD symptoms, directly facilitated by the psychiatric assistance dog's trained tasks and in many cases also emergent from the human-animal bond itself. A greater sense of safety and calm, improved peace of mind, augmented sense of self-worth, and increased emotional reserves ultimately translated to

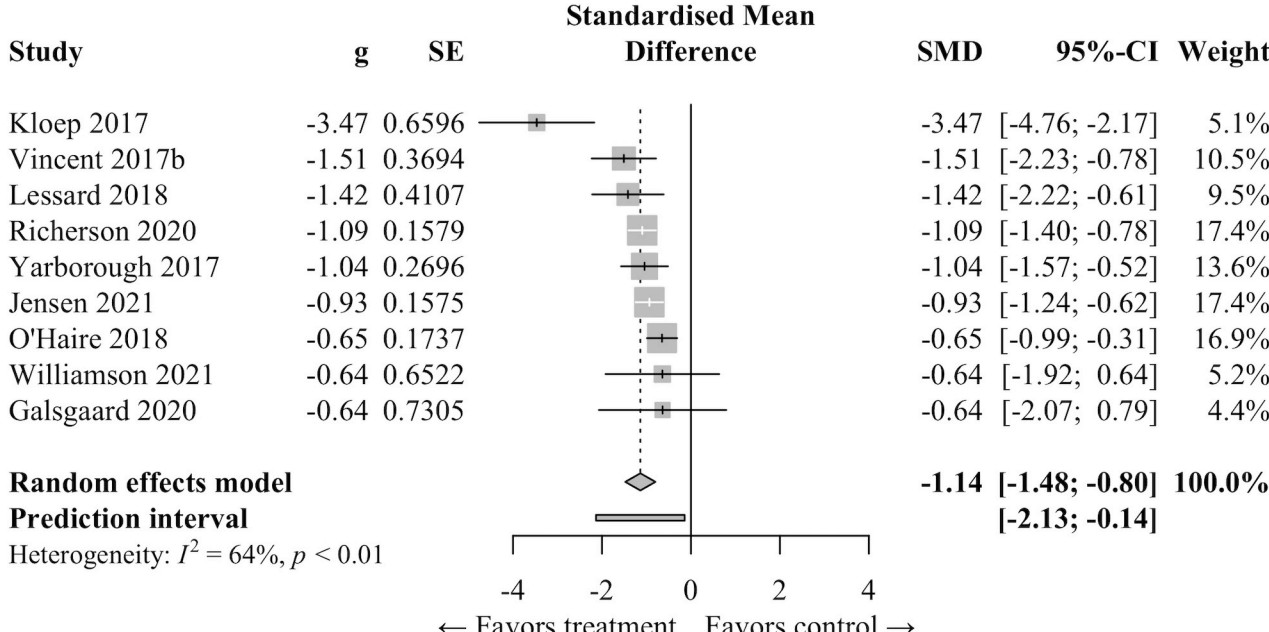

**Fig 6. Forest plot on impact of service dog placement on PTSD checklist for DSM-5 (PCL-5) scores for veterans with PTSD.**

**Table 4. PCL-5 scores for studies included in meta-analysis.**

| Study | No Dog | | | Dog | | Δ M | Rigor |
|---|---|---|---|---|---|---|---|
| | Control | M (SD) | n | M (SD) | n | | |
| Kloep 2017* | Pre | 52.00 (12.5) | 12 | 15.00 (7.5) | 12 | -37.00 | 93% |
| Vincent 2017b* | Pre | 61.00 (10.0) | 19 | 40.00 (16.5) | 19 | -21.00 | 53% |
| Lessard 2018* | Pre | 56.00 (12.0) | 15 | 35.00 (16.5) | 15 | -21.00 | 87% |
| Richerson 2020 | Pre | 48.33 (15.7) | 97 | 31.66 (14.6) | 88 | -16.67 | 93% |
| Yarborough 2017* | WL | 56.00 (14.5) | 51 | 38.00 (22.0) | 22 | -18.00 | 93% |
| Jensen 2021 | WL | 58.97 (13.0) | 74 | 44.34 (17.1) | 112 | -14.63 | 93% |
| O'Haire 2018* | WL | 56.00 (14.5) | 66 | 46.00 (16.0) | 74 | -10.00 | 100% |
| Williamson 2021 | Pre | 60.20 (14.5) | 5 | 48.10 (19.2) | 5 | -12.10 | 82% |
| Galsgaard 2020* | Pre | 45.00 (15.5) | 4 | 34.00 (14.5) | 4 | -11.00 | 55% |
| **Summary** | | **54.54** | **343** | **39.41** | **351** | **-15.13** | |

Notes. Order corresponds to Fig 6, i.e., according to effect size (Hedge's g).

* Scores crosswalked from PCL-C, -M, or -S for comparability to PCL-5. WL Veterans on a waitlist to receive a service dog. Pre Baseline timepoint prior to placement with a service dog.

A change in PCL-5 score of 10 or more indicates clinically meaningful change; a score of 33 or higher is considered indicative of a PTSD diagnosis.

transformative change for many veterans. The topic of treatment participation came up in two articles, both concluding that the assistance dog appeared to improve compliance in medical treatment [44, 45]. Finally, multiple quoted participants went so far as to say that partnership with their assistance dog may have prevented them from dying by suicide [42, 44, 46, 63–65].

Changes in certain areas of physical health were evident as well: improvements in sleep quality was a theme among first and second order constructs for most articles [42, 44, 45, 63, 65–70], as were decreases in prescription medication use and possible improvements substance use [42, 44, 45, 54, 63, 66–68, 71]. One article also identified improvements in physical safety thanks to trained tasks to support balance and picking up of dropped items, although these types of tasks are generally considered more typical of a mobility rather than psychiatric assistance dog [67].

The impacts on the individual were not universally positive; while not present in every article, most spoke to some of the challenges inherent to the psychiatric assistance dog acquisition process and partnership [42, 45, 63, 65, 67–69]. Challenges in the acquisition process were most evident in one article, which was unique in that a subset of participants were interviewed both before and after partnership with the assistance dog [63]. The process of training with an organization to acquire an assistance dog can be demanding and stressful, and beneficial changes are not necessarily immediate. Once partnered the most common challenges pertaining to this theme related to the dog's ongoing care and anticipation of grief at the prospect of the assistance dog passing away in the future. Multiple articles suggest that these challenges may be possible to mitigate through adequate preparation and adherence to high standards on the part of the assistance dog organization [44, 45, 63, 68], with participants in two of these studies specifically recommending selecting an organization accredited by ADI.

*2. Impact beyond the individual: building relationships & connection*

Further analysis of first and second order constructs led to the identification of a second theme. The "impact beyond the individual" speaks to the ways in which partnership with an assistance dog, and indeed even participation in an assistance dog intervention in the first place, facilitates human connection for the veteran handler. This theme was absent from only one article, which may be explained by the fact that their paper focused on the impact of assistance dog placements on substance use rather than interpersonal interactions [71].

Isolation can be prevalent among veterans with military-connected PTSD, and this stood out as a noteworthy factor motivating participants to apply for a psychiatric assistance dog [42, 63–66, 70]. The application for an assistance dog and the subsequent training and partnership process mark the first occasion in which the veteran begins to form new connections and relationships by virtue of participation in an assistance dog intervention. The importance of this moment is highlighted in multiple articles, all of which speak the importance of high standards and quality on the part of the assistance dog organization at this critical juncture [42, 44, 45, 68].

Once partnered psychiatric assistance dogs act as a "social bridge," influencing existing relationships and facilitating the formation of new connections [42, 44–46, 62–65, 67, 68, 72]. This appears to be an overall net positive effect in both domains. The assistance dog partnership can facilitate the veteran reconnecting or repairing existing relationships. It can also promote increased social and community engagement outside of existing social networks by enabling the veteran to enter public spaces or participate in activities that were previously inaccessible due to their PTSD symptoms. Several quoted individuals felt that partnership with a psychiatric assistance dog was a factor in helping them gain or keep employment [42, 64, 68]. The role of the assistance dog as a social bridge can in some cases be quite literal: the dog's presence creates a safe topic of conversation that facilitates emotionally safe, positive connections with strangers, thereby closing the existing gap in communication. Likewise, development of a healthy relationship schema with the assistance dog appeared, for some participants, to prompt beneficial improvements in their existing human relationships.

Beyond benefits, there are nuances to the assistance dog's influence on the veteran's social connections that bring new challenges unique to the assistance dog partnership [42, 44–46, 62, 63, 65, 67–69, 72]. For existing relationships this is most evident within the family system, which was most comprehensively described by the two studies that uniquely included veterans' partners among study participants [62, 63]. The addition of an assistance dog appears to introduce increased relational load; there is a new burden of care for a living being, and navigating the complex shift in dynamics for human caretakers is not always a smooth process. Likewise, the presence of an assistance dog alongside the veteran can result in new challenges in public. These challenges generally do not appear to relate to the dog's behavior but rather unwanted attention, stigma, or disrespect from the public, encountering poorly behaved fraudulent assistance dogs, and in some cases experiencing public access denials.

## Discussion

The practice and study of psychiatric assistance dog placements for military veterans with PTSD have risen dramatically in recent years. This systematic literature review sought to synthesize existing literature on the subject, identifying 41 articles (including 29 peer-reviewed publications and 12 unpublished dissertations) meeting inclusion criteria, encompassing 1,765 veteran participants. Overall, research on this topic is extremely recent: the oldest article was published only 8 years ago in 2014, and 100% of peer-reviewed articles were published within the last 5 years. This growth aligns with the increase in psychiatric assistance dogs as a percentage of the assistance dog population overall, from 17% of placements in 2000–2002 to over 30% starting in 2010 [17].

A small number (7) of articles explored primarily non-veteran participants, speaking to the depth and complexity of the community involved in successful assistance dog partnership, beyond simply the handler-canine partners to include mental health providers, assistance dog organizations or trainers, family members, businesses, and members of the public. Only one article examined the welfare of the assistance dogs themselves, pointing to an important gap for future research. The remaining 34 articles with primarily veteran participants were

included in specific aims analyses to summarize placement characteristics, assess methodologi-cal rigor, and summarize outcomes.

## Characteristics

Study participants were primarily male, white, veterans of the United States Army, and an average of 42 years old. All assistance dog organizations were nonprofit and most were ADI-accredited. Most dogs were Labrador Retrievers originating from shelters; purpose-bred assis-tance dogs were considerably underrepresented in the sample compared to the industry overall [5, 17]. Salient placement and demographic details were often not reported, interfering with our ability to discern comparability and quality of the psychiatric assistance dog intervention across studies. Many different factors have the potential to influence outcomes, and these details are particularly important to report given the inherent complexity of animal-assisted interventions (which include two unique and complicated organisms) and the lack of stan-dardization and oversight within the assistance dog industry [17, 73]. Future research should endeavor to meet established recommendations for methodological rigor (e.g., [74–77]) including reporting detailed participant, organization, and canine demographics (i.e., partici-pant age, gender, race and ethnicity, trauma origin, concurrent treatment, comorbid diagno-ses, and military branch or branches; organization name, accreditation status, human-canine pairing format; and canine breed, origin, age, trained tasks, and training format). Additionally, given that the population in question is veterans with PTSD, studies should consider referring to the Common Data Elements for PTSD Research for additional elements to report [78, 79]. Not only are these elements important to permit interpretation of results, they are also crucial to ensure the possibility of future study replication (e.g., [80]).

Demographically, we were interested in whether United States veteran participants are rep-resentative of the population of United States veterans more broadly; unfortunately, statistics on veteran demographics globally were unavailable. Although participants included in the United States studies appear somewhat representative of the population of United States veter-ans overall, they do not appear to be fully representative of the population of treatment-seeking veterans with PTSD more broadly, and PTSD is known to affect Black and Hispanic veterans at elevated rates [81]. White veterans are greatly overrepresented in the overall sample (78%, compared to 46% of treatment-seeking veterans with PTSD in the United States) while Black veterans in particular appear highly underrepresented (7%, compared to 20%) [82]. These could be indicators that research studies are recruiting veterans from different demographic characteristics disproportionately or may be a sign that the complementary intervention of a psychiatric assistance dog is not equally accessible to all veterans irrespective of demographic characteristics. This finding signals an important area for future research to explore, particu-larly given the known racial and ethnic disparities in healthcare in PTSD diagnosis and treat-ment in the United States [83, 84].

## Methodological rigor

Neither methodological rigor nor proportion of significant findings differed between peer-reviewed publications and theses, suggesting less concern for a file drawer effect in this particu-lar sample. Methodological rigor varied widely across studies, with larger studies meeting higher rigor criteria. This large range reflects known challenges in the human-animal interac-tion field more broadly [80]. Building upon the study characteristics summarized above, the most salient area for growth in the methodological rigor was the description of placement characteristics. In general, most studies incorporated control conditions of waitlist (56%) or

pre-post (28%), permitting between-group comparisons and strengthening confidence in interpretability of findings.

Ultimately, at this time causality cannot be inferred even for statistically significant findings, due to the lack of RCTs overall, with the sole exception of the congressionally-mandated VA study. However, the comparison condition employed in this study was placement with an emotional support dog rather than a true standard care, or "no dog" comparison, and the emotional support dogs in question were atypical due to their high level of training and evaluation, all of which likely introduced significant confounding variables. Therefore, to understand whether psychiatric assistance dogs can be considered an evidence-based complementary intervention for veterans with PTSD, additional research with a RCT design employing a standard care comparison condition is needed.

## Outcomes

To assess outcomes of psychiatric assistance dog placements for military-connected PTSD, we conducted both a meta-analysis (quantitative) and meta-synthesis (qualitative). Findings support the conclusion that placement with a psychiatric assistance dog is associated with a meaningful decrease in PTSD symptoms but should not be considered a "cure" or standalone treatment. Based on our meta-analysis, placement with an assistance dog had a large effect on PTSD symptoms: it was significantly associated with a 1.14 standard deviation decrease in PCL-5 score. While the mean score at follow-up remained above the diagnostic cutoff score of 33 (mean of 39.41, possible range 0–80), the mean change of −15.13 does approach the VA's calculated mean change of −15.8 for the intervention to be considered "cost-effective" [85], and exceeds the threshold of −10 for clinically significant improvement.

Several additional pieces of information should be considered in interpreting this finding. One, the range of post-placement ("dog") average time points for the included studies is from 3–22 months, whereas on average assistance dog placements can last until the dog is 10 ½ years of age [86]. It is unclear at this point how PTSD symptoms may evolve over a longer period of time–whether they continue to decrease, level off, or fluctuate in other ways. The effect of assistance dog loss, whether through retirement or death, is also unknown and of potential concern [86]. Additionally, outcomes resulting from placement with a psychiatric assistance dog likely vary widely from veteran to veteran. Notably, the sole RCT reported that 31% of veterans in the assistance dog group no longer met clinical criteria for PTSD at 15 months post-placement based on the gold-standard CAPS-5 clinical assessment [18]. An important area for future research will be to further elucidate the mechanisms and moderators influencing the efficacy of this complementary intervention over time.

Improvements in PTSD symptoms may be driven in part by the assistance dog's trained tasks. Notably, research indicates that these tasks help with some but not all clusters of PTSD symptoms (e.g., hypervigilance, unwanted upsetting memories, and startle reactions; but not recall of trauma or participation in risky/destructive behavior; [6]). This finding may provide insight as to why PTSD symptoms remain above the diagnostic cutoff score despite significant improvements. Additionally, there was variation in the distribution of tasks reported in the literature, which could reflect variation in the tasks being offered by assistance dog organizations. In particular, although calming and social greeting tasks have been found to be among the most important for veterans with PTSD [6, 87], they were mentioned in relatively few articles. Furthermore, although nightmare interruption was among the most commonly cited tasks, at this time it is unclear how service dogs might reliably recognize a nightmare. There is currently minimal evidence that trauma-related nightmares are externally observable, although of the available research it appears possible that respirator events or limb movement

could serve as cues in at least some cases [88–90]. While a few studies have begun to examine the associations between trained tasks and outcomes, their underlying processes and importance as a potential mechanism for improvement will continue to be a key area for future research.

The complex and strong human-animal bond between handler and canine may have an even greater influence on outcomes than the assistance dog's trained tasks, although the nuances of this dynamic are poorly-defined at this stage. Findings to date indicate potential associations between outcomes and attachment style, the dog's "untrained" behaviors, strength of bond, and canine personality. The emergent relationship may primarily be a healing one, but it is also accompanied by new challenges and vulnerabilities (e.g., the added burden of care for the dog and anticipation of grief upon loss). Centering our understanding of this bond within the context of an existing theoretical framework is likely to help clarify the mechanisms at play; most frequently, articles suggest attachment theory, social support theory, and the biophilia hypothesis as potential candidates [91–93]. Overall, the relative paucity of findings in this domain highlights an interesting area for future research to explore.

In addition to clear areas of gains, the synthesis of research also highlights clear areas of null findings, which are essential to establish discriminant validity. The impacts of assistance dogs for PTSD are most salient for mental health, but do not appear to substantially change physical health. This may have blunted overall effects for global quality of life instruments and could be due to the fact that psychiatric assistance dogs are trained to impact mental (and not physical) health symptoms, or could be a result of ceiling effects–if an individual was already in good physical health, there may have been little room for significant improvement. Contradictory results were also evident in some areas including sleep, medication use, and substance use: while veterans self-report experiencing improvements in these domains, null findings from objective measures did not support these conclusions, highlighting a need for further research and the possibility that perceived and objective benefits may not align. Finally, while not researched in this population to date, service dog handlers may experience health benefits (e.g., improved cardiovascular health; [94]) equivalent to that observed in pet dog owners.

## Future directions

There are indications that psychiatric assistance dogs may positively influence mental health treatment participation (or at the very least, that they do not negatively impact treatment participation), which may alleviate concerns in the mental health community that partnership with a psychiatric assistance dog would lead to decreased treatment participation [33, 95]. These are encouraging findings; however, drawing conclusions would be premature due to the aforementioned underreporting of concurrent treatment participation in most articles. Moreover, the mental health community and government agencies such as the VA will need to contend with the inherent discrepancy between their apparent desired goals (e.g., reduced health care costs vs. increased access to and participation in concurrent treatments), which cannot both be achieved simultaneously. Additionally, neither quantitative nor qualitative research has, at this time, addressed the question of whether working with a psychiatric assistance animal constitutes a maladaptive safety behavior for the veteran handler [33, 96, 97]. In answering this question, empirical investigation into specific trained tasks is warranted to understand whether or not their use is counter-therapeutically aligned with avoidance symptoms.

Future research will also need to more thoroughly examine the impact of psychiatric assistance dogs on suicidality. Given that rates of suicide among veterans are substantially higher than among civilians [1], the scarcity of data on this topic–measured by only one of 20 quantitative and mixed methods studies–was unexpected. While qualitative studies point to

potentially important improvements in suicidality, additional research could provide empirical evidence as to what is and is not realistic to expect from partnership with a psychiatric assistance dog.

Finally, an important and largely unaddressed area for future research will be to explore the influence of psychiatric assistance dog placement on social health from diversity, equity, inclusion, and belonging as well as trauma-informed lenses. The loss of social participation due to isolation, common among veterans with PTSD, is clearly profoundly detrimental: both to the individual's own well-being as well as to the community which is unable to benefit from their unique perspective and potential contributions. Moreover, as members of the community of people with invisible disabilities (i.e., disabilities that cannot be easily identified visually), veterans with PTSD may experience distress not only as a direct result of their symptoms, but also due to stigma and invalidation from the community–including from within the disabled community itself [98]. Researchers must take into account the fact that for these veterans, partnership with a psychiatric assistance dog is effectively a disclosure of disability (and possibly also veteran) status, and can therefore lead to increased discrimination. In summary, it is critical for future research on this subject to retain a broad lens, looking at *and beyond* the individual to include the community as a whole.

## Limitations

Although we endeavored to cast a wide net in our search for eligible articles by including 11 databases (including grey literature), it is possible that we missed articles on this subject if they did not appear in the search results. Additionally, our criteria included publication in English which had the potential to result in selection bias; however, no articles were excluded only on the basis of this criterion. Finally, due to the wide variety in study designs and measures used to assess quantitative outcomes, only a small subset of articles reporting results using the same measure could ultimately be included in the meta-analysis.

## Conclusion

Increasingly prevalent research on psychiatric assistance dogs for military veterans with PTSD provides support for the positive impact of this complementary intervention on PTSD symptom severity. Meaningful improvements in adjacent domains span both mental and social health. Possible mechanisms include the assistance dog's trained tasks and the complex bond shared by handler and canine. Future research should endeavor to report detailed participant, organization, and canine demographic information. Key opportunities include examining the welfare of the assistance dog themselves, the accessibility of the psychiatric assistance dog intervention, the mechanisms and moderators underlying influencing the intervention's efficacy, and research to understand impacts on areas beyond PTSD symptoms such as suicidality and treatment participation. Ultimately, a randomized controlled trial with a standard care, "no dog" comparison condition will be needed to permit causal inferences as to the true impact of psychiatric assistance dog placement for veterans with military-connected posttraumatic stress disorder.

## Supporting information

**S1 Table. Database search vocabulary and syntax.** ti Title. ab Abstract. Exact search syntax was adjusted based on the vocabulary for each database.
(DOCX)

**S2 Table. Methodological rigor scoring questions.**
(DOCX)

**S3 Table. Characteristics of service dog organizations.** Ordered by most recent to least recent within each group.—Not reported. ADI Assistance Dogs International. AKC CGC American Kennel Club Canine Good Citizen Test. d Day. w Week. m Month. y Year. [a]Status not reported in publication, but presumed based on other publications with same organization.
(DOCX)

**S4 Table. Characteristics of service dogs.** Ordered by most recent to least recent within each group.—Not reported. LR Labrador Retriever. GR Golden Retriever. LX Labrador Retriever Mix. GSD German Shepherd Dog. X Mixed breed. V Veteran (with organizational guidance). [a]—Trained tasks are not reported; N Trained tasks are reported, but individual PTSD-specific tasks are not described; Y Trained tasks are reported, and PTSD-specific trained tasks are described.
(DOCX)

**S1 Checklist. Prisma checklist.**
(PDF)

## Author Contributions

**Conceptualization:** Sarah C. Leighton, Leanne O. Nieforth, Marguerite E. O'Haire.

**Data curation:** Sarah C. Leighton, Leanne O. Nieforth.

**Formal analysis:** Sarah C. Leighton, Leanne O. Nieforth.

**Methodology:** Sarah C. Leighton, Leanne O. Nieforth.

**Project administration:** Sarah C. Leighton, Marguerite E. O'Haire.

**Supervision:** Marguerite E. O'Haire.

**Visualization:** Sarah C. Leighton, Leanne O. Nieforth.

**Writing – original draft:** Sarah C. Leighton.

**Writing – review & editing:** Sarah C. Leighton, Leanne O. Nieforth, Marguerite E. O'Haire.

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
