## [Decision Letter · Decision Letter 0]

8 Jul 2022

PONE-D-22-11188Assistance dogs for military veterans with PTSD: A systematic review, meta-analysis, and meta-synthesisPLOS ONE

Dear Dr. Leighton,

Thank you for submitting your manuscript to PLOS ONE. After careful consideration, we feel that it has merit but does not fully meet PLOS ONE’s publication criteria as it currently stands. Therefore, we invite you to submit a revised version of the manuscript that addresses the points raised during the review process.

Thank you for submitting this systematic review. 

I have a few comments that you may consider:

Abstract: please add number of studies that are included in the meta-analysis

It took me some time to understand table 1 - maybe you should show the studies on veterans before showing the studies on non-veteran participants?

L 415: the RCT study - please specify  that the first line measures PTSD severity and add % of veterans no longer meeting clinical criteria for PTSD in the control group.

L 527: Meta-analysis - did you conduct any sensitivity analysis? one study has a much larger effect size and you might want to look into this - if the study has any specific characteristics which can explain this. if you leave this study out are the results still significant?

L 622 Mixed methods articles: I do not think that this section adds much so I would suggest adding the results of these two studies to the quantitave and qualitative result sections 

Countries: when reading he paper it sounds as if all studies are US - but here are studies from other countries. please make this more clear e.g. in the paragraph starting in L 678

L 750 Discussion please add references when referring to theoretical frameworks

I have not been able to access supplementary files so could not check them

We look forward to receiving your revised manuscript.

Kind regards,

Maiken Pontoppidan

Academic Editor

PLOS ONE

Journal Requirements:

Reviewers' comments:

Reviewer's Responses to Questions

**Comments to the Author**

1. Is the manuscript technically sound, and do the data support the conclusions?

Reviewer #1: Yes

Reviewer #2: Yes

2. Has the statistical analysis been performed appropriately and rigorously? 

Reviewer #1: I Don't Know

Reviewer #2: Yes

3. Have the authors made all data underlying the findings in their manuscript fully available?

Reviewer #1: Yes

Reviewer #2: Yes

4. Is the manuscript presented in an intelligible fashion and written in standard English?

Reviewer #1: Yes

Reviewer #2: Yes

5. Review Comments to the Author

Reviewer #1: I am assuming that the use of substance use disorder, substance abuse, etc. are used to reflect the language used in the specific articles.

Line 112 - should it be identified what was found?

Line 313 - So how do these studies account for confounding factors, such as other treatment modalities. And if they do not, does this need to be made clearer?

May want to note in the abstract the original 432 studies are not US specific; it sounds like they are from the previous sentence.

Reviewer #2: This is an excellent meta-analysis in every aspect. I look forward to referring colleagues and students to it as a "must read" in this field. I have only a few quibbles.

The "space creation" training standard for PTSD service dogs is accurately described as a common and popular training target, but it is also one that has been opposed by the PTSD treatment community as counter-therapeutically aligned with avoidance symptoms. An effort was aimed at removing this item from the ADI standards. As these are now published only in summary form, it is difficult to determine the current status of this standard. As this paper functions as well as a scoping review of the field, some reference to the controversy would be appropriate. And it would harmonize with the welcome attention the authors draw to "trained" and "untrained" behaviors and their potential impacts on outcomes.

In the service of scientific rigor, the purported "nightmare interruption" function of PTSD service dogs deserves serious review. Notwithstanding the compelling anecdotal reports, it is worth considering how a service dog might be trained to perform this function. First, trauma-related nightmares occur unpredictably during sleep in the home environment, precluding a role for a professional trainer. Second, there is no evidence that trauma-related nightmares are systematically associated with detectable motor or verbal behaviors that could cue a dog to intervene.

Lastly, while there are currently few or no data supporting physical health benefits associated with service dog partnering in veterans, two large epidemiological studies by Mubanga found 20-30% reductions in cardiovascular and all-cause mortality in Swedish civilians who owned dogs versus those that did not. While these papers do not involve veterans with PTSD, the latter suffer elevated rates of cardiovascular disease. In serving as a scoping review as well as a meta-analysis, the paper could be justified in touching on this consequential and under-researched area of inquiry.

6. PLOS authors have the option to publish the peer review history of their article (what does this mean?). If published, this will include your full peer review and any attached files.

Reviewer #1: No

Reviewer #2: No

---

## [Author Response · Author response to Decision Letter 0]

13 Jul 2022

We have also uploaded the following in an easier-to-read format in the file labeled Response to Reviewers.

Dear Reviewers, 

Thank you very much for your time and expertise in reviewing our manuscript. We are grateful for your suggestions which we believe have strengthened the quality of our manuscript. Please find the itemized responses to each of your comments below, in addition to an updated manuscript using track changes. 

Editorial Recommendations: 

Abstract: please add number of studies that are included in the meta-analysis

We appreciate this recommendation and have added this information: “Meta-analysis (9 articles) revealed that partnership with an assistance dog had a clinically meaningful, significant, and large effect on PTSD severity scores (g=−1.129; p<0.0001).” (line 38)

It took me some time to understand table 1 - maybe you should show the studies on veterans before showing the studies on non-veteran participants?

We have edited this section and the title of Table 1 to simplify and provide greater clarity about why the non-veteran participants were included separately, prior to the main analyses. These studies provide background context and set the stage for understanding the subsequent veteran-focused studies.

The start of this section now reads “Seven articles were excluded from specific aims analyses, as the participants in these studies were not themselves veterans (33–39). Excluded articles, summarized in Table 1. The participants in these articles were partners and family members, the staff involved in the training and placement of the dog, the mental health professionals providing care, and the assistance dogs themselves.” (lines 249-260)

The title of Table 1 is now “Non-veteran participant studies: Participants, design, and outcomes.”

L 415: the RCT study - please specify that the first line measures PTSD severity and add % of veterans no longer meeting clinical criteria for PTSD in the control group.

We appreciate this recommendation and have added this information. The manuscript now reads: “The sole randomized controlled trial (RCT, conducted by the VA), which was also the only study to report PTSD severity findings for a clinical measure (i.e., the CAPS-5), found no significant between-group difference for veterans paired with assistance dogs compared to emotional support dogs; however, 32% of the assistance dog group and 26% of the control group no longer met clinical criteria for PTSD at 15 months.” (lines 449-453)

L 527: Meta-analysis - did you conduct any sensitivity analysis? one study has a much larger effect size and you might want to look into this - if the study has any specific characteristics which can explain this. if you leave this study out are the results still significant?

Thank you for this question. We ran a sensitivity analysis removing Kloep 2017, and the results are still significant (p < 0.0001). The small sample size of this study, which may be contributing to the large effect size, is accounted for in the model with a weight of 5.1% (see rightmost column of Figure 6). 

L 622 Mixed methods articles: I do not think that this section adds much so I would suggest adding the results of these two studies to the quantitative and qualitative result sections.

Thank you for this recommendation, we have made this change and adjusted those sections accordingly.

Countries: when reading the paper it sounds as if all studies are US - but here are studies from other countries. Please make this more clear e.g. in the paragraph starting in L 678

We appreciate this observation and have added clarifying language: 

“Demographically, we were interested in whether United States veteran participants are representative of the population of United States veterans more broadly; unfortunately, statistics on veteran demographics globally were unavailable. Although participants included in the United States studies appear somewhat representative of the population of United States veterans overall, they do not appear to be fully representative…” (lines 737-743)

“This finding signals an important area for future research to explore, particularly given the known racial and ethnic disparities in healthcare in PTSD diagnosis and treatment in the United States.” (lines 750-752)

L 750 Discussion please add references when referring to theoretical frameworks

We appreciate this recommendation and have added the following references for each of the theoretical frameworks to line 819:

91. Lawrence EA. The sacred bee, the filthy pig, and the bat out of hell: Animal symbolism as cognitive biophilia. The biophilia hypothesis. 1993;301–41. 

92. Beck AM, Katcher AH. Future directions in human-animal bond research. Am Behav Sci. 2003;47(1):79–93. 

93. Bowlby J. Attachment and loss: retrospect and prospect. Am J Orthopsychiatry. 1982;52(4):664.

I have not been able to access supplementary files so could not check them

Please let us know if there is anything we can do to assist.

Reviewer #1: 

I am assuming that the use of substance use disorder, substance abuse, etc. are used to reflect the

language used in the specific articles.

Thank you for drawing our attention to this language use. We have made revisions for consistency and to align with recommended non-stigmatizing language (National Institute on Drug Abuse, 2021), using “substance use” except where referring specifically to the diagnosis of substance use disorder. 

Line 112 - should it be identified what was found?

The outcomes of this study are identified in our results section, thus are not included in the introduction.

Line 313 - So how do these studies account for confounding factors, such as other treatment modalities. And if they do not, does this need to be made clearer?

Thank you for this question. We have added information to clarify this point: “The vast majority of studies did not account for other treatment modalities as confounding factors; however, some indicated equivalence across groups at baseline (12%, n=4), uniform participation in psychiatric treatment as an inclusion requirement for the study (6%, n=2) or the assistance dog program (12%, n=4), or reported it as an outcome (n=3 quantitative, n=2 qualitative).” (lines 328-332) 

Additionally, outcomes for articles reporting results relating to treatment participation are reported in lines 521-527.

May want to note in the abstract the original 432 studies are not US specific; it sounds like they are from the previous sentence.

Thank you for drawing our attention to this error. The 19% number referenced in the abstract is in fact global and not US-specific. We have made this correction: “Psychiatric assistance dogs for military veterans with posttraumatic stress disorder (PTSD) currently make up over 19% of assistance dog partnerships globally.” (line 25)

Reviewer #2: 

This is an excellent meta-analysis in every aspect. I look forward to referring colleagues and students to

it as a "must read" in this field. I have only a few quibbles.

Thank you for the very kind feedback.

The "space creation" training standard for PTSD service dogs is accurately described as a common and popular training target, but it is also one that has been opposed by the PTSD treatment community as counter-therapeutically aligned with avoidance symptoms. An effort was aimed at removing this item from the ADI standards. As these are now published only in summary form, it is difficult to determine the current status of this standard. As this paper functions as well as a scoping review of the field, some reference to the controversy would be appropriate. And it would harmonize with the welcome attention the authors draw to "trained" and "untrained" behaviors and their potential impacts on outcomes.

We are appreciative of these insights and have expanded our discussion of safety behaviors accordingly. While we have personal knowledge of the ADI standards development discussions surrounding these tasks, we were unable to identify appropriate citations to discuss this at greater length. 

The manuscript now reads: “In answering this question, empirical investigation into specific trained tasks is warranted to understand whether or not their use is counter-therapeutically aligned with avoidance symptoms.” (lines 847-849)

In the service of scientific rigor, the purported "nightmare interruption" function of PTSD service dogs deserves serious review. Notwithstanding the compelling anecdotal reports, it is worth considering how a service dog might be trained to perform this function. First, trauma-related nightmares occur unpredictably during sleep in the home environment, precluding a role for a professional trainer. Second, there is no evidence that trauma-related nightmares are systematically associated with detectable motor or verbal behaviors that could cue a dog to intervene.

Thank you for highlighting this important topic. We have made revisions recommending further inquiry into this task, in alignment with this comment. 

The manuscript now reads: “Furthermore, although nightmare interruption was among the most commonly cited tasks, at this time it is unclear how service dogs might reliably recognize a nightmare. There is currently minimal evidence that trauma-related nightmares are externally observable, although of the available research it appears possible that respirator events or limb movement could serve as cues in at least some cases (88–90). While a few studies have begun to examine the associations between trained tasks and outcomes, their underlying processes and importance as a potential mechanism for improvement will continue to be a key area for future research.” (lines 802-809)

88. Mellman TA, Kulick-Bell R, Ashlock LE, Nolan B. Sleep events among veterans with combat-related posttraumatic stress disorder. Am J Psychiatry. 1995 Jan;152(1):110–5. 

89. Paul F, Alpers GW, Reinhard I, Schredl M. Nightmares do result in psychophysiological arousal: A multimeasure ambulatory assessment study. Psychophysiology. 2019;56(7):e13366. 

90. Phelps AJ, Kanaan RAA, Worsnop C, Redston S, Ralph N, Forbes D. An Ambulatory Polysomnography Study of the Post-traumatic Nightmares of Post-traumatic Stress Disorder. Sleep. 2018 Jan 1;41(1).

Lastly, while there are currently few or no data supporting physical health benefits associated with service dog partnering in veterans, two large epidemiological studies by Mubanga found 20-30% reductions in cardiovascular and all-cause mortality in Swedish civilians who owned dogs versus those that did not. While these papers do not involve veterans with PTSD, the later suffer elevated rates of cardiovascular disease. In serving as a scoping review as well as a meta-analysis, the paper could be justified in touching on this consequential and under-researched area of inquiry.

We appreciate this point and have added to our discussion accordingly: “Finally, while not researched in this population to date, service dog handlers may experience health benefits (e.g., improved cardiovascular health; 94) equivalent to that observed in pet dog owners.” (lines 831-833)

94. Mubanga M, Byberg L, Egenvall A, Ingelsson E, Fall T. Dog ownership and survival after a major cardiovascular event: a register-based prospective study. Circulation: Cardiovascular Quality and Outcomes. 2019;12(10):e005342.

---

## [Decision Letter · Decision Letter 1]

23 Aug 2022

PONE-D-22-11188R1Assistance dogs for military veterans with PTSD: A systematic review, meta-analysis, and meta-synthesisPLOS ONE

Dear Dr. Leighton,

Thank you for submitting your manuscript to PLOS ONE. After careful consideration, we feel that it has merit but does not fully meet PLOS ONE’s publication criteria as it currently stands. Therefore, we invite you to submit a revised version of the manuscript that addresses the points raised during the review process.

We look forward to receiving your revised manuscript.

Kind regards,

Guangyu Tong

Academic Editor

PLOS ONE

Journal Requirements:

Additional Editor Comments (if provided):

Please address the remaining comments accordingly.

Reviewers' comments:

Reviewer's Responses to Questions

**Comments to the Author**

1. If the authors have adequately addressed your comments raised in a previous round of review and you feel that this manuscript is now acceptable for publication, you may indicate that here to bypass the “Comments to the Author” section, enter your conflict of interest statement in the “Confidential to Editor” section, and submit your "Accept" recommendation.

Reviewer #2: All comments have been addressed

2. Is the manuscript technically sound, and do the data support the conclusions?

Reviewer #2: (No Response)

3. Has the statistical analysis been performed appropriately and rigorously? 

Reviewer #2: (No Response)

4. Have the authors made all data underlying the findings in their manuscript fully available?

Reviewer #2: (No Response)

5. Is the manuscript presented in an intelligible fashion and written in standard English?

Reviewer #2: (No Response)

6. Review Comments to the Author

Reviewer #2: (No Response)

7. PLOS authors have the option to publish the peer review history of their article (what does this mean?). If published, this will include your full peer review and any attached files.

Reviewer #2: No

---

## [Author Response · Author response to Decision Letter 1]

6 Sep 2022

Dear Reviewers, 

Thank you very much for your time and expertise in reviewing our manuscript.

Reviewer #1: 

- We were not provided with any comments from Reviewer #1. We thank them again for their time and expertise in reviewing our manuscript and hope that we were successful in addressing all of their previous comments.

Reviewer #2: 

"All comments have been addressed."

- We are pleased to have addressed all of Reviewer #2’s comments and thank them again for their valuable feedback, which we believe has strengthened the quality of our manuscript.

---

## [Editor Report · Decision Letter 2]

8 Sep 2022

Assistance dogs for military veterans with PTSD: A systematic review, meta-analysis, and meta-synthesis

PONE-D-22-11188R2

Dear Dr. Leighton,

We’re pleased to inform you that your manuscript has been judged scientifically suitable for publication and will be formally accepted for publication once it meets all outstanding technical requirements.

Kind regards,

Guangyu Tong

Academic Editor

PLOS ONE

---

## [Editor Report · Acceptance letter]

12 Sep 2022

PONE-D-22-11188R2 

Assistance dogs for military veterans with PTSD: A systematic review, meta-analysis, and meta-synthesis 

Dear Dr. Leighton:

I'm pleased to inform you that your manuscript has been deemed suitable for publication in PLOS ONE. Congratulations! Your manuscript is now with our production department. 

Kind regards, 

on behalf of

Dr. Guangyu Tong 

Academic Editor

PLOS ONE